# A Half-Space Stochastic Projected Gradient Method for Group Sparsity Regularization

## Abstract

Optimizing with group sparsity is significant in enhancing model interpretability in machining learning applications, *e.g.*, feature selection, compressed sensing and model compression. However, for large-scale stochastic training problems, effective group sparsity exploration are typically hard to achieve. Particularly, the state-of-the-art stochastic optimization algorithms usually generate merely dense solutions. To overcome this shortage, we propose a stochastic method—Half-space Stochastic Projected Gradient (HSPG) method to search solutions of high group sparsity while maintain the convergence. Initialized by a simple Prox-SG Step, the HSPG method relies on a novel Half-Space Step to substantially boost the sparsity level. Numerically, HSPG demonstrates its superiority in deep neural networks, *e.g.*, VGG16, ResNet18 and MobileNetV1, by computing solutions of higher group sparsity, competitive objective values and generalization accuracy.

## 1 Introduction

In many recent machine learning optimization tasks, researchers not only focus on finding solutions with small prediction/generalization error but also concentrate on improving the interpretation of model by filtering out redundant parameters and achieving slimmer model architectures. One technique to achieve the above goal is by augmenting the sparsity-inducing regularization terms to the raw objective functions to generate sparse solutions (including numerous zero elements). The popular $\ell_1$-regularization promotes the sparsity of solutions by element-wise penalizing the optimization variables. However, in many practical applications, there exist additional constraints on variables such that the zero coefficients are often not randomly distributed but tend to be clustered into varying more sophisticated sparsity structures, *e.g.*, disjoint and overlapping groups and hierarchy (Yuan & Lin, 2006; Huang et al., 2010; 2009). As the most important and natural form of structured sparsity, the disjoint group-sparsity regularization, which assumes the pre-specified disjoint blocks of variables are selected (non-zero variables) or ignored (zero variables) simultaneously (Bach et al., 2012), serves as a momentous role in general structured sparsity learning tasks since other instances such as overlapping group and hierarchical sparsity are typically solved by converting into the equivalent disjoint group versions via introducing latent variables (Bach et al., 2012), and has found numerous applications in computer vision (Elhamifar et al., 2012), signal processing (Chen & Selesnick, 2014), medical imaging (Liu et al., 2018), and deep learning (Scardapane et al., 2017), especially on the model compression of deep neural networks, where the group sparsity[1] is leveraged to remove redundant entire hidden structures directly.

**Problem Setting.** We study the disjoint group sparsity regularization problem which can be typically formulated as the mixed $\ell_1/\ell_p$-regularization problem, and pay special attention to the most popular and widely used instance $p$ as 2 (Bach et al., 2012; Halabi et al., 2018),

$$\underset{\boldsymbol{x} \in \mathbb{R}^n}{\text{minimize}} \left\{ \Psi(\boldsymbol{x}) \overset{\text{def}}{=} f(\boldsymbol{x}) + \lambda \Omega(\boldsymbol{x}) = \frac{1}{N} \sum_{i=1}^{N} f_i(\boldsymbol{x}) + \lambda \sum_{g \in \mathcal{G}} \|[\boldsymbol{x}]_g\| \right\}, \tag{1}$$

where $\lambda > 0$ is a weighting factor, $\|\cdot\|$ denotes $\ell_2$-norm, $f(\boldsymbol{x})$ is the average of numerous $N$ continuously differentiable instance functions $f_i : \mathbb{R}^n \to \mathbb{R}$, such as the loss functions measuring the deviation from the observations in various data fitting problems, $\Omega(\boldsymbol{x})$ is the so-called mixed $\ell_1/\ell_2$

---

[1]Group sparsity is defined as # of zero groups, where a zero group means all its variables are exact zeros.

norm, $\mathcal{G}$ is a prescribed fixed partition of index set $\mathcal{I} = \{1, 2, \cdots, n\}$, wherein each component $g \in \mathcal{G}$ indexes a group of variables upon the perspective of applications. Theoretically, a larger $\lambda$ typically results in a higher group sparsity while sacrifices more on the bias of model estimation, hence $\lambda$ needs to be carefully fine-tuned to achieve both low $f$ and high group-sparse solutions.

**Literature Review.** Problem (1) has been well studied in deterministic optimization with various algorithms that are capable of returning solutions with both low objective value and high group sparsity under proper $\lambda$ (Yuan & Lin, 2006; Roth & Fischer, 2008; Huang et al., 2011; Ndiaye et al., 2017). Proximal methods are classical approaches to solve the structured non-smooth optimization (1), including the popular proximal gradient method (Prox-FG) which only uses the first-order derivative information. When $N$ is huge, stochastic methods become ubiquitous to operate on a small subset to avoid the costly evaluation over all instances in deterministic methods for large-scale problems. Proximal stochastic gradient method (Prox-SG) (Duchi & Singer, 2009) is the natural stochastic extension of Prox-FG. Regularized dual-averaging method (RDA) (Xiao, 2010; Yang et al., 2010) is proposed by extending the dual averaging scheme in (Nesterov, 2009). To improve the convergence rate, there exists a set of incremental gradient methods inspired by SAG (Roux et al., 2012) to utilizes the average of accumulated past gradients. For example, proximal stochastic variance-reduced gradient method (Prox-SVRG) (Xiao & Zhang, 2014) and proximal spider (Prox-Spider) (Zhang & Xiao, 2019) are developed to adopt multi-stage schemes based on the well-known variance reduction technique SVRG proposed in (Johnson & Zhang, 2013) and Spider developed in (Fang et al., 2018) respectively. SAGA (Defazio et al., 2014) stands as the midpoint between SAG and Prox-SVRG.

Compared to deterministic methods, the studies of mixed $\ell_1/\ell_2$-regularization (1) in stochastic field become somewhat rare and limited. Prox-SG, RDA, Prox-SVRG, Prox-Spider and SAGA are valuable state-of-the-art stochastic algorithms for solving problem (1) but with apparent weakness. Particularly, these existing stochastic algorithms typically meet difficulties to achieve both decent convergence and effective group sparsity identification simultaneously (e.g., small function values but merely dense solutions), because of the randomness and the limited sparsity-promotion mechanisms. In depth, Prox-SG, RDA, Prox-SVRG, Prox-Spider and SAGA derive from proximal gradient method to utilize the proximal operator to produce group of zero variables. Such operator is generic to extensive non-smooth problems, consequently perhaps not sufficiently insightful if the target problems possess certain properties, *e.g.*, the group sparsity structure as problem (1). In fact, in convex setting, the proximal operator suffers from variance of gradient estimate; and in non-convex setting, especially deep learning, the discreet step size (learning rate) further deteriorates its effectiveness on the group sparsity promotion, as will show in Section 2 that the projection region vanishes rapidly except RDA. RDA has superiority on finding manifold structure to others (Lee & Wright, 2012), but inferiority on the objective convergence. Besides, the variance reduction techniques are typically required to measure over a huge mini-batch data points in both theory and practice which is probably prohibitive for large-scale problems, and have been observed as sometimes noneffective for deep learning applications (Defazio & Bottou, 2019). On the other hand, to introduce sparsity, there exist heuristic weight pruning methods (Li et al., 2016; Luo et al., 2017), whereas they commonly do not equip with theoretical guarantee, so that easily diverge and hurt generalization accuracy.

**Our Contributions.** Half-Space Stochastic Projected Gradient (HSPG) method overcomes the limitations of the existing stochastic algorithms on the group sparsity identification, while maintains comparable convergence characteristics. While the main-stream works on (group) sparsity have focused on using proximal operators of regularization, our method is unique and fresh in enforcing group sparsity more effectively by leveraging half-space structure and is well supported by the theoretical analysis and empirical evaluations. We now summarize our contributions as follows.

- *Algorithmic Design:* We propose the HSPG to solve the disjoint group sparsity regularized problem as (1). Initialized with a Prox-SG Step for seeking a close-enough but perhaps dense solution estimate, the algorithmic framework relies on a novel Half-Space Step to exploit group sparse patterns. We delicately design the Half-Space Step with the following main features: *(i)* it utilizes previous iterate as the normal direction to construct a reduced space consisting of a set of half-spaces and the origin; *(ii)* a new group projection operator maps groups of variables onto zero if they fall out of the constructed reduced space to identify group sparsity considerably more effectively than the proximal operator; and *(iii)* with proper step size, the Half-Space Step enjoys the sufficient decrease property, and achieves progress to optimum in both theory and practice.

- *Theoretical Guarantee:* We provide the convergence guarantees of HSPG. Moreover, we prove HSPG has looser requirements to identify the sparsity pattern than Prox-SG, revealing its superiority

on the group sparsity exploration. Particularly, for the sparsity pattern identification, the required distance to the optimal solution $\boldsymbol{x}^*$ of HSPG is better than the distance required by Prox-SG.

- *Numerical Experiments:* Experimentally, HSPG outperforms the state-of-the-art methods in the aspect of the group sparsity exploration, and achieves competitive objective value convergence and runtime in both convex and non-convex problems. In the popular deep learning tasks, HSPG usually computes the solutions with multiple times higher group sparsity and similar generalization performance on unseen testing data than those generated by the competitors, which may be further used to construct smaller and more efficient network architectures.

## 2  THE HSPG METHOD

We state the Half-Space Stochastic Projected Gradient (HSPG) method in Algorithm 1. In general, it contains two stages: Initialization Stage and Group-Sparsity Stage. The first Initialization Stage employs Prox-SG Step (Algorithm 2) to search for a close-enough but usually non-sparse solution estimate. Then the second and fundamental stage proceeds Half-Space Step (Algorithm 3) started with the non-sparse solution estimate to effectively exploit the group sparsity within a sequence of reduced spaces, and converges to the group-sparse solutions with theoretical convergence property.

---

**Algorithm 1** Outline of HSPG for solving (1).

1: **Input:** $\boldsymbol{x}_0 \in \mathbb{R}^n$, $\alpha_0 \in (0, 1)$, $\epsilon \in [0, 1)$, and $N_{\mathcal{P}} \in \mathbb{Z}^+$.
2: **for** $k = 0, 1, 2, \ldots$ **do**
3:     **if** $k < N_{\mathcal{P}}$ **then**
4:         Compute $\boldsymbol{x}_{k+1} \leftarrow$ Prox-SG$(\boldsymbol{x}_k, \alpha_k)$ by Algorithm 2.
5:     **else**
6:         Compute $\boldsymbol{x}_{k+1} \leftarrow$ Half-Space$(\boldsymbol{x}_k, \alpha_k, \epsilon)$ by Algorithm 3.
7:     Update $\alpha_{k+1}$.

---

**Algorithm 2** Prox-SG Step.

1: **Input:** Current iterate $\boldsymbol{x}_k$, and step size $\alpha_k$.
2: Compute the stochastic gradient of $f$ on mini-batch $\mathcal{B}_k$

$$\nabla f_{\mathcal{B}_k}(\boldsymbol{x}_k) \leftarrow \frac{1}{|\mathcal{B}_k|} \sum_{i \in \mathcal{B}_k} \nabla f_i(\boldsymbol{x}_k). \tag{2}$$

3: **Return** $\boldsymbol{x}_{k+1} \leftarrow \text{Prox}_{\alpha_k \lambda \Omega(\cdot)} \left( \boldsymbol{x}_k - \alpha_k \nabla f_{\mathcal{B}_k}(\boldsymbol{x}_k) \right)$.

---

**Initialization Stage.** The Initialization Stage performs the vanilla proximal stochastic gradient method (Prox-SG, Algorithm 2) to approach the solution of (1). At $k$th iteration, a mini-batch $\mathcal{B}_k$ is sampled to generate an unbiased estimator of the full gradient of $f$ (line 2, Algorithm 2) to compute a trial iterate $\widehat{\boldsymbol{x}}_{k+1} := \boldsymbol{x}_k - \alpha_k \nabla f_{\mathcal{B}_k}(\boldsymbol{x}_k)$, where $\alpha_k$ is the step size, and $f_{\mathcal{B}_k}$ is the average of the instance functions $f_i$ cross $\mathcal{B}_k$. The next iterate $\boldsymbol{x}_{k+1}$ is then updated based on the proximal mapping

$$\boldsymbol{x}_{k+1} = \text{Prox}_{\alpha_k \lambda \Omega(\cdot)}(\hat{\boldsymbol{x}}_{k+1}) = \underset{\boldsymbol{x} \in \mathbb{R}^n}{\arg\min} \ \frac{1}{2\alpha_k} \|\boldsymbol{x} - \hat{\boldsymbol{x}}_{k+1}\|^2 + \lambda \Omega(\boldsymbol{x}), \tag{3}$$

where the regularization term $\Omega(\boldsymbol{x})$ is defined in (1). Notice that the above subproblem (3) has a closed-form solution, where for each $g \in \mathcal{G}$, we have

$$[\boldsymbol{x}_{k+1}]_g = \max\left\{0, 1 - \alpha_k \lambda / \|[\widehat{\boldsymbol{x}}_{k+1}]_g\|\right\} \cdot [\widehat{\boldsymbol{x}}_{k+1}]_g. \tag{4}$$

In HSPG, the Initialization Stage proceeds Prox-SG Step $N_{\mathcal{P}}$ times as a localization mechanism to seek an estimation which is close enough to a solution of problem (1), where $N_{\mathcal{P}} := \min\{k : k \in \mathbb{Z}^+, \|\boldsymbol{x}_k - \boldsymbol{x}^*\| \leq R/2\}$ associated with a positive constant $R$ related to the optima, see (23) in Appendix C. In practice, although the close-enough requirement is perhaps hard to be verified, we empirically suggest to keep running the Prox-SG Step until observing some stage-switch signal by testing on the stationarity of objective values, norm of (sub)gradient or validation accuracy similarly to (Zhang et al., 2020). However, the Initialization Stage alone is *insufficient* to exploit the group

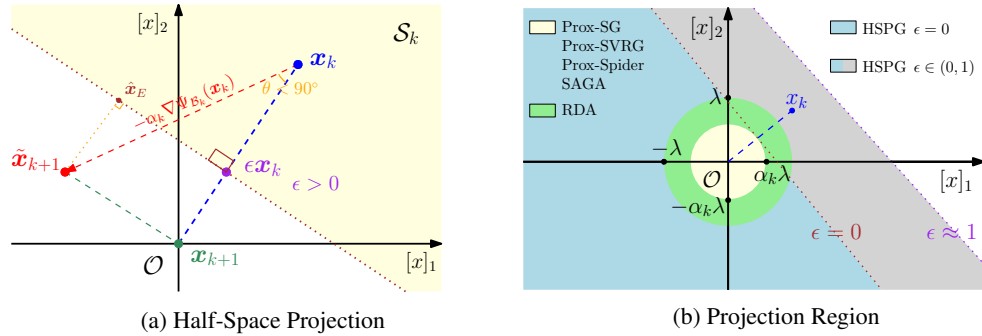

(a) Half-Space Projection

(b) Projection Region

Figure 1: Illustration of Half-Space Step with projection in (9), where $\mathcal{G} = \{\{1, 2\}\}$.

sparsity structure, *i.e.*, the computed solution estimate is typically dense, due to the randomness and the moderate truncation mechanism of proximal operator constrained in its projection region, *i.e.*, the trial iterate $[\widehat{\boldsymbol{x}}_{k+1}]_g$ is projected to zero only if it falls into an $\ell_2$-ball centered at the origin with radius $\alpha_k \lambda$ by (4). Our remedy is to incorporate it with the following Half-Space Step, which exhibits an effective sparsity promotion mechanism while still remains the convergent property.

---

**Algorithm 3** Half-Space Step

---

1: **Input:** Current iterate $\boldsymbol{x}_k$, step size $\alpha_k$, and $\epsilon$.
2: Compute the stochastic gradient of $\Psi$ on $\mathcal{I}^{\neq 0}(\boldsymbol{x}_k)$ by mini-batch $\mathcal{B}_k$

$$[\nabla\Psi_{\mathcal{B}_k}(\boldsymbol{x}_k)]_{\mathcal{I}^{\neq 0}(\boldsymbol{x}_k)} \leftarrow \frac{1}{|\mathcal{B}_k|} \sum_{i \in \mathcal{B}_k} [\nabla\Psi_i(\boldsymbol{x}_k)]_{\mathcal{I}^{\neq 0}(\boldsymbol{x}_k)} \tag{5}$$

3: Compute $[\tilde{\boldsymbol{x}}_{k+1}]_{\mathcal{I}^{\neq 0}(\boldsymbol{x}_k)} \leftarrow [\boldsymbol{x}_k - \alpha_k \nabla\Psi_{\mathcal{B}_k}(\boldsymbol{x}_k)]_{\mathcal{I}^{\neq 0}(\boldsymbol{x}_k)}$ and $[\tilde{\boldsymbol{x}}_{k+1}]_{\mathcal{I}^0(\boldsymbol{x}_k)} \leftarrow 0$.
4: **for** each group $g$ in $\mathcal{I}^{\neq 0}(\boldsymbol{x}_k)$ **do**
5:     **if** $[\tilde{\boldsymbol{x}}_{k+1}]_g^\top [\boldsymbol{x}_k]_g < \epsilon \|[\boldsymbol{x}_k]_g\|^2$ **then**
6:         $[\tilde{\boldsymbol{x}}_{k+1}]_g \leftarrow 0$.
7: **Return** $\boldsymbol{x}_{k+1} \leftarrow \tilde{\boldsymbol{x}}_{k+1}$.

---

**Group-Sparsity Stage.** The Group-Sparsity Stage is designed to effectively determine the groups of zero variables and capitalize convergence characteristic, which is in sharp contrast to other heuristic aggressive weight pruning methods but typically lacking theoretical guarantee (Li et al., 2016; Luo et al., 2017). The underlying intuition of its atomic Half-Space Step (Algorithm 3) is to project $[\boldsymbol{x}_k]_g$ to zero only if $-[\boldsymbol{x}_k]_g$ serves as a descent step to $\Psi(\boldsymbol{x}_k)$, *i.e.*, $-[\boldsymbol{x}_k]_g^\top [\nabla\Psi(\boldsymbol{x}_k))]_g < 0$, hence updating $[\boldsymbol{x}_{k+1}]_g \leftarrow [\boldsymbol{x}_k]_g - [\boldsymbol{x}_k]_g = 0$ still results in some progress to the optimality. Before introducing that, we first define the following index sets for any $\boldsymbol{x} \in \mathbb{R}^n$:

$$\mathcal{I}^0(\boldsymbol{x}) := \{g : g \in \mathcal{G}, [\boldsymbol{x}]_g = 0\} \text{ and } \mathcal{I}^{\neq 0}(\boldsymbol{x}) := \{g : g \in \mathcal{G}, [\boldsymbol{x}]_g \neq 0\}, \tag{6}$$

where $\mathcal{I}^0(\boldsymbol{x})$ represents the indices of groups of zero variables at $\boldsymbol{x}$, and $\mathcal{I}^{\neq 0}(\boldsymbol{x})$ indexes the groups of nonzero variables at $\boldsymbol{x}$. To proceed, we further define an artificial set that $\boldsymbol{x}$ lies in:

$$\mathcal{S}(\boldsymbol{x}) := \left\{ \boldsymbol{z} \in \mathbb{R}^n : [\boldsymbol{z}]_g = \boldsymbol{0} \text{ if } g \in \mathcal{I}^0(\boldsymbol{x}), \text{and } [\boldsymbol{z}]_g^\top [\boldsymbol{x}]_g \geq \epsilon \|[\boldsymbol{x}]_g\|^2 \text{ if } g \in \mathcal{I}^{\neq 0}(\boldsymbol{x}) \right\} \bigcup \{\boldsymbol{0}\}, \tag{7}$$

which consists of half-spaces and the origin. Here the parameter $\epsilon > 0$ controls the grey region presented in Figure 1b, and the exact way to set $\epsilon$ will be discussed in Section 4 and Appendix. Hence, $\boldsymbol{x}$ inhabits $\mathcal{S}(\boldsymbol{x}_k)$, *i.e.*, $\boldsymbol{x} \in \mathcal{S}(\boldsymbol{x}_k)$, only if: *(i)* $[\boldsymbol{x}]_g$ lies in the upper half-space for all $g \in \mathcal{I}^{\neq 0}(\boldsymbol{x}_k)$ for some prescribed $\epsilon \in [0, 1)$ as shown in Figure 1a; and *(ii)* $[\boldsymbol{x}]_g$ equals to zero for all $g \in \mathcal{I}^0(\boldsymbol{x}_k)$. The fundamental assumption for Half-Space Step to success is that: the Initialization Stage has produced a (possibly *non-sparse*) solution estimate $\boldsymbol{x}_k$ nearby a group sparse solution $\boldsymbol{x}^*$ of problem (1), *i.e.*, the optimal distance $\|\boldsymbol{x}_k - \boldsymbol{x}^*\|$ is sufficiently small. As seen in Appendix, it further indicates that the group sparse optimal solution $\boldsymbol{x}^*$ inhabits $\mathcal{S}_k := \mathcal{S}(\boldsymbol{x}_k)$, which implies that

$\mathcal{S}_k$ has already covered the group-support of $\boldsymbol{x}^*$, *i.e.*, $\mathcal{I}^{\neq 0}(\boldsymbol{x}^*) \subseteq \mathcal{I}^{\neq 0}(\boldsymbol{x}_k)$. Our goal now becomes minimizing $\Psi(\boldsymbol{x})$ over $\mathcal{S}_k$ to identify the remaining groups of zero variables, *i.e.*, $\mathcal{I}^0(\boldsymbol{x}^*)/\mathcal{I}^0(\boldsymbol{x}_k)$, which is formulated as the following smooth optimization problem:

$$\boldsymbol{x}_{k+1} = \underset{\boldsymbol{x} \in \mathcal{S}_k}{\arg\min} \ \Psi(\boldsymbol{x}) = f(\boldsymbol{x}) + \lambda\Omega(\boldsymbol{x}). \tag{8}$$

By the definition of $\mathcal{S}_k$, $[\boldsymbol{x}]_{\mathcal{I}^0(\boldsymbol{x}_k)} \equiv \boldsymbol{0}$ are constrained as fixed during Algorithm 3 proceeding, and only the entries in $\mathcal{I}^{\neq 0}(\boldsymbol{x}_k)$ are allowed to move. Hence $\Psi(\boldsymbol{x})$ is smooth on $\mathcal{S}_k$, and (8) is a reduced space optimization problem. A standard way to solve problem (8) would be the stochastic gradient descent equipped with Euclidean projection (Nocedal & Wright, 2006). However, such a projected method rarely produces zero (group) variables as the dense $\hat{x}_E$ illustrated in Figure 1a. To address it, we introduce a novel projection operator to effectively conduct group projection as follows.

As stated in Algorithm 3, we first approximate the gradient of $\Psi$ on the free variables in $\mathcal{I}^{\neq 0}(\boldsymbol{x}_k)$ by $[\nabla\Psi_{\mathcal{B}_k}(\boldsymbol{x}_k)]_{\mathcal{I}^{\neq 0}(\boldsymbol{x}_k)}$ (line 2, Algorithm 3), then employ SGD to compute a trial point $\widetilde{\boldsymbol{x}}_{k+1}$ (line 3, Algorithm 3) which is passed into a new projection operator $\text{Proj}_{\mathcal{S}_k}(\cdot)$ defined as

$$\left[\text{Proj}_{\mathcal{S}_k}(\boldsymbol{z})\right]_g := \begin{cases} [\boldsymbol{z}]_g & \text{if } [\boldsymbol{z}]_g^\top [\boldsymbol{x}_k]_g \geq \epsilon \|[\boldsymbol{x}_k]_g\|^2, \\ 0 & \text{otherwise.} \end{cases} \tag{9}$$

The above projector of form (9) is not the standard Euclidean projection operator in most cases[2], but still satisfies the following two advantages: *(i)* the actual search direction $\boldsymbol{d}_k := (\text{Proj}_{\mathcal{S}_k}(\widetilde{\boldsymbol{x}}_{k+1}) - \boldsymbol{x}_k)/\alpha_k$ performs as a descent direction to $\Psi_{\mathcal{B}_k}(\boldsymbol{x}_k) := f_{\mathcal{B}_k}(\boldsymbol{x}_k) + \lambda\Omega(\boldsymbol{x}_k)$, *i.e.*, $[\boldsymbol{d}_k]_g^\top[\nabla\Psi_{\mathcal{B}_k}(\boldsymbol{x}_k))]_g < 0$ as $\theta < 90°$ in Figure 1a, then the progress to the optimum is made via the sufficient decrease property as drawn in Lemma 1; and *(ii)* effectively project groups of variables to zero simultaneously if the inner product of corresponding entries is sufficiently small. In contrast, the Euclidean projection operator is far away effective to promote group sparsity, as the Euclidean projected point $\hat{\boldsymbol{x}}_E \neq 0$ versus $\boldsymbol{x}_{k+1} = \text{Proj}_{\mathcal{S}_k}(\widetilde{\boldsymbol{x}}_{k+1}) = 0$ shown in Figure 1a.

**Lemma 1.** *Algorithm 3 yields the next iterate $\boldsymbol{x}_{k+1}$ as $Proj_{\mathcal{S}_k}(\boldsymbol{x}_k - \alpha_k\nabla\Psi_{\mathcal{B}_k}(\boldsymbol{x}_k))$, then the search direction $\boldsymbol{d}_k := (\boldsymbol{x}_{k+1} - \boldsymbol{x}_k)/\alpha_k$ is a descent direction for $\Psi_{\mathcal{B}_k}(\boldsymbol{x}_k)$, i.e., $\boldsymbol{d}_k^\top\nabla\Psi_{\mathcal{B}_k}(\boldsymbol{x}_k) < 0$. Moreover, letting $L$ be the Lipschitz constant for $\nabla\Psi_{\mathcal{B}_k}$ on the feasible domain, and $\hat{\mathcal{G}}_k := \mathcal{I}^{\neq 0}(\boldsymbol{x}_k) \bigcap \mathcal{I}^0(\boldsymbol{x}_{k+1})$ and $\tilde{\mathcal{G}}_k := \mathcal{I}^{\neq 0}(\boldsymbol{x}_k) \bigcap \mathcal{I}^{\neq 0}(\boldsymbol{x}_{k+1})$ be the sets of groups which projects or not onto zero, we have*

$$\Psi_{\mathcal{B}_k}(\boldsymbol{x}_{k+1}) \leq \Psi_{\mathcal{B}_k}(\boldsymbol{x}_k) - \left(\alpha_k - \frac{\alpha_k^2 L}{2}\right) \sum_{g \in \tilde{\mathcal{G}}_k} \|[\nabla\Psi_{\mathcal{B}_k}(\boldsymbol{x}_k)]_g\|^2 - \left(\frac{1-\epsilon}{\alpha_k} - \frac{L}{2}\right) \sum_{g \in \hat{\mathcal{G}}_k} \|[\boldsymbol{x}_k]_g\|^2. \tag{10}$$

We then intuitively illustrate the strength of HSPG on group sparsity exploration. In fact, the half-space projection (9) is a more effective sparsity promotion mechanism compared to the existing methods. Particularly, it benefits from a much larger projection region to map a reference point $\hat{\boldsymbol{x}}_{k+1} := \boldsymbol{x}_k - \alpha_k\nabla f_{\mathcal{B}_k}(\boldsymbol{x}_k)$ or its variants to zero. As the 2D case described in Figure 1b, the projection regions of Prox-SG, Prox-SVRG, Prox-Spider and SAGA are $\ell_2$-balls with radius as $\alpha_k\lambda$. In stochastic learning, especially deep learning tasks, the step size $\alpha_k$ is usually selected around $10^{-3}$ to $10^{-4}$ or even smaller for convergence. Together with the common setting of $\lambda \ll 1$, their projection regions would vanish rapidly, resulting in the difficulties to produce group sparsity. As a sharp contrast, even though $\alpha_k\lambda$ is near zero, the projection region of HSPG $\{\boldsymbol{x} : \boldsymbol{x}_k^\top\boldsymbol{x} < (\alpha_k\lambda + \epsilon\|\boldsymbol{x}_k\|)\|\boldsymbol{x}_k\|\}$ (seen in Appendix) is still an open half-space which contains those $\ell_2$ balls as well as RDA's if $\epsilon$ is large enough. Moreover, the positive control parameter $\epsilon$ adjusts the level of aggressiveness of group sparsity promotion (9), *i.e.*, the larger the more aggressive, and meanwhile maintains the progress to the optimality by Lemma 1. In practice, proper fine tuning $\epsilon$ is sometimes required to achieve both group sparsity enhancement and sufficient decrease on objective value as will see in Section 4.

**Intuition of Two-Stage Method:** To end this section, we discuss the advantage of designing such two stage schema rather than an adaptive switch back and forth between the Prox-SG Step and Half-Space Step based on some evaluation switching criteria, as many multi-step deterministic optimization algorithms (Chen et al., 2017). In fact, we numerically observed that switching back to the Prox-SG Step consistently deteriorate the progress of group sparsity exploration by Half-Space Step while without obvious gain on convergence. Such regression on group sparsity by the Prox-SG Step

---

[2]Unless $\Omega(\boldsymbol{x})$ is $\|\boldsymbol{x}\|_1$ where each $g \in \mathcal{G}$ is singleton, then $\mathcal{S}_k$ becomes an orthant face (Chen et al., 2020).

is less attractive in realistic applications, *e.g.*, model compression, where people usually possess heavy models of high generalization accuracy ahead and want to filter out the redundancy effectively. Therefore, in term of the ease of application, we end at organizing Prox-SG Step and Half-Space Step as such a two-stage schema, controlled by a switching hypermeter $N_{\mathcal{P}}$. In theory, we require $N_{\mathcal{P}}$ sufficiently large to let the initial iterate of Half-Space Step be close enough to the local minimizer as shown in Section 3. In practice, HSPG is sensitive to the choice of $N_{\mathcal{P}}$ at early iterations, *i.e.*, switching to Half-Space Step too early may result accuracy loss. But such sensitivity vanishes rapidly if switching to Half-Space Step after some acceptable evaluation switching criteria.

## 3 CONVERGENCE ANALYSIS

In this section, we give the convergence guarantee of our HSPG. Towards that end, we make the following widely used assumption in optimization literature (Xiao & Zhang, 2014; Yang et al., 2019) and active set identification analysis of regularization problem (Nutini et al., 2019; Chen et al., 2018).

**Assumption 1.** *Each $f_i : \mathbb{R}^n \to \mathbb{R}$, for $i = 1, 2, \cdots, N$, is differentiable and bounded below. Their gradients $\nabla f_i(\boldsymbol{x})$ are Lipschitz continuous, and let $L$ be the shared Lipschitz constant.*

**Assumption 2.** *The least and the largest $\ell_2$-norm of non-zero groups in $\boldsymbol{x}^*$ are lower and upper bounded by some constants, i.e., $0 < 2\delta_1 := \min_{g \in \mathcal{I}^{\neq 0}(\boldsymbol{x}^*)} \|[\boldsymbol{x}^*]_g\|$ and $0 < 2\delta_2 := \max_{g \in \mathcal{I}^{\neq 0}(\boldsymbol{x}^*)} \|[\boldsymbol{x}^*]_g\|$. Moreover, we request a common strict complementarity on any $\mathcal{B}$, i.e., $0 < 2\delta_3 := \min_{g \in \mathcal{I}^0(\boldsymbol{x}^*)} (\lambda - \|[\nabla f_{\mathcal{B}}(\boldsymbol{x}^*)]_g\|)$ for regularization optimization.*

**Notations:** Let $\boldsymbol{x}^*$ be a local minimizer of problem (1) with group sparsity property, $\Psi^*$ be the local minimum value corresponding to $\boldsymbol{x}^*$, and $\{\boldsymbol{x}_k\}_{k=0}^\infty$ be the iterates generated from Algorithm 1. Denote the gradient mapping of $\Psi(\boldsymbol{x})$ and its estimator on mini-batch $\mathcal{B}$ as $\boldsymbol{\xi}_\eta(\boldsymbol{x}) := \frac{1}{\eta} \left( \boldsymbol{x} - \text{Prox}_{\eta\lambda\Omega(\cdot)}(\boldsymbol{x} - \eta\nabla f(\boldsymbol{x})) \right)$ and $\boldsymbol{\xi}_{\eta,\mathcal{B}}(\boldsymbol{x}) := \frac{1}{\eta} \left( \boldsymbol{x} - \text{Prox}_{\eta\lambda\Omega(\cdot)}(\boldsymbol{x} - \eta\nabla f_{\mathcal{B}}(\boldsymbol{x})) \right)$ respectively. We say $\tilde{\boldsymbol{x}}$ a stationary point of $\Psi(\boldsymbol{x})$ if $\boldsymbol{\xi}_\eta(\tilde{\boldsymbol{x}}) = 0$. To be simple, let $\widetilde{\mathcal{X}}$ be a neighbor of $\boldsymbol{x}^*$ as $\widetilde{\mathcal{X}} := \{\boldsymbol{x} : \|\boldsymbol{x} - \boldsymbol{x}^*\| \leq R\}$ with $R$ as a positive constant related to $\delta_1, \delta_2$ and $\epsilon$ (see (23) in Appendix C), and $M$ be the supremum of $\|\partial\Psi(\boldsymbol{x})\|$ on the compact set $\widetilde{\mathcal{X}}$.

**Remark:** Assumption 1 implies that $\nabla f_{\mathcal{B}}(\boldsymbol{x})$ measured on mini-batch $\mathcal{B}$ is Lipschitz continuous on $\mathbb{R}^n$ with the same Lipschitz constant $L$, while $\nabla\Psi_{\mathcal{B}}(\boldsymbol{x})$ is not as shown in Appendix. However, the Lipschitz continuity of $\nabla\Psi_{\mathcal{B}}(\boldsymbol{x})$ still holds on $\mathcal{X} = \{\boldsymbol{x} : \|[\boldsymbol{x}]_g\| \geq \delta_1$ for each $g \in \mathcal{G}\}$ by excluding a $\ell_2$-ball centered at the origin with radius $\delta_1$ from $\mathbb{R}^n$. For simplicity, let $\nabla\Psi_{\mathcal{B}}(\boldsymbol{x})$ share the same Lipschitz constant $L$ on $\mathcal{X}$ with $\nabla f_{\mathcal{B}}(\boldsymbol{x})$, since we can always select the bigger value as their shared Lipschitz constant. Now, we state the first main theorem of HSPG.

**Theorem 1.** *Suppose $f$ is convex on $\widetilde{\mathcal{X}}$, $\epsilon \in \left[0, \min\left\{\frac{\delta_1^2}{\delta_2}, \frac{2\delta_1 - R}{2\delta_2 + R}\right\}\right)$, $\|\boldsymbol{x}_K - \boldsymbol{x}^*\| \leq \frac{R}{2}$ for $K \geq N_{\mathcal{P}}$. Set $k := K + t$, $(t \in \mathbb{Z}^+)$. Then for any $\tau \in (0, 1)$, there exist step size $\alpha_k = \mathcal{O}(\frac{1}{\sqrt{Nt}}) \in \left(0, \min\left\{\frac{2(1-\epsilon)}{L}, \frac{1}{L}, \frac{2\delta_1 - R - \epsilon(2\delta_2 + R)}{M}\right\}\right)$, and mini-batch size $|\mathcal{B}_k| = \mathcal{O}(t) \leq N - \frac{N}{2M}$, such that $\{\boldsymbol{x}_k\}$ converges to some stationary point in expectation with probability at least $1 - \tau$, i.e., $\mathbb{P}(\lim_{k \to \infty} \mathbb{E}[\|\boldsymbol{\xi}_{\alpha_k, \mathcal{B}_k}(\boldsymbol{x}_k)\|] = 0) \geq 1 - \tau$.*

**Remark:** Theorem 1 only requires local convexity of $f$ on a neighborhood $\widetilde{\mathcal{X}}$ of $\boldsymbol{x}^*$ while itself can be non-convex in general. This local convexity assumption appears in many non-convex analysis, such as: tensor decomposition (Ge et al., 2015) and shallow neural networks (Zhong et al., 2017). Theorem 1 implies that if the $K$th iterate locates close enough to $\boldsymbol{x}^*$, the step size $\alpha_k$ and mini-batch size $|\mathcal{B}_k|$ is set as above, (it further indicates $\boldsymbol{x}^*$ inhabits the $\{\mathcal{S}_k\}_{k \geq K}$ of all subsequent iterates updated by Half-Space Step with high probability in Appendix), then the Half-Space Step in Algorithm 3 guarantees the convergence to the stationary point. The $\mathcal{O}(t)$ mini-batch size is commonly used in the analysis of stochastic algorithms, *e.g.*, Adam and Yogi (Zaheer et al., 2018). Later based on numerical results in Section 4, we observe that a much weaker increasing or even constant mini-batch size is sufficient. In fact, experiments show that practically, a reasonably large mini-batch size can work well if the variance is not large. Although the assumption $\|\boldsymbol{x}_K - \boldsymbol{x}^*\| < R/2$ is hard to be verified in practice, setting $N_{\mathcal{P}}$ large enough usually performs quite well.

We then reveal the sparsity identification guarantee of HSPG as stated in Theorem 2.

**Theorem 2.** *If $k \geq N_{\mathcal{P}}$ and $\|x_k - x^*\| \leq \frac{2\alpha_k \delta_3}{1-\epsilon+\alpha_k L}$, then HSPG yields $\mathcal{I}^0(x^*) \subseteq \mathcal{I}^0(x_{k+1})$.*

**Remark:** Theorem 2 shows that when $x_k$ is in the $\ell_2$-ball centered at $x^*$ with radius $\frac{2\alpha_k \delta_3}{1-\epsilon+\alpha_k L}$, HSPG identifies the optimal sparsity pattern, *i.e.*, $\mathcal{I}^0(x^*) \subseteq \mathcal{I}^0(x_{k+1})$. In contrast, to identify the sparsity pattern, Prox-SG requires the iterates to fall into the $\ell_2$-ball centered at $x^*$ with radius $\alpha_k \delta_3$ (Nutini et al., 2019). Since $\alpha_k \leq 1/L$ and $\epsilon \in [0,1)$, then $\frac{2\alpha_k \delta_3}{1-\epsilon+\alpha_k L} \geq \alpha_k \delta_3$ implies that the $\ell_2$-ball of HSPG contains the $\ell_2$-ball of Prox-SG, *i.e.*, HSPG has a stronger performance in sparsity pattern identification. Therefore, Theorem 2 reveals a better sparsity identification property of HSPG than Prox-SG, and no similar results exist for other methods to our knowledge.

**The Initialization Stage Selection:** To satisfy the pre-requirement of convergence of Half-Space Step as Theorem 1, *i.e.*, initial iterate close enough to $x^*$, there exists several proper candidates *e.g.*, Prox-SG, Prox-SVRG and SAGA to form as the Initialization Stage. Considering the tradeoff between computational efficiency and theoretical convergence, our default setting is to select Prox-SG. Although Prox-SVRG/SAGA may have better theoretical convergence property than Prox-SG, they require higher time and space complexity to compute or estimate full gradient on a huge mini-batch or store previous gradient, which may be prohibitive for large-scale training especially when the memory is often limited. Besides, it is well noticed that SVRG does not work as desired on the popular non-convex deep learning applications (Defazio & Bottou, 2019; Chen et al., 2020). In contrast, Prox-SG is efficient and can also achieves the good initialization assumption in Theorem 1, *i.e.*, $\|x_{N_{\mathcal{P}}} - x^*\| \leq R/2$, in the manner of high probability via performing sufficiently many times, as revealed in Appendix C.4 by leveraging related literature (Rosasco et al., 2019) associated with an additional strongly convex assumption. However, one should notice that Prox-SG does not guarantee any group sparsity property of $x_{N_{\mathcal{P}}}$ due to the limited projection region and randomness.

**Remark**: We emphasize that this paper focuses on improving the group sparsity identification, which is rarely explored and also a key indicator of success for structured sparsity regularization problem. Meanwhile, we would like to point out improving the convergence rate has been very well explored in a series of literatures (Reddi et al., 2016; Li & Li, 2018), but out of our main consideration.

## 4 NUMERICAL EXPERIMENTS

In this section, we present results of several benchmark numerical experiments in deep neural networks to illustrate the superiority of HSPG than other related algorithms on group sparsity exploration and the comparable convergence. Besides, two extensible convex experiments are conducted in Appendix to empirically demonstrate the validness and superiority of the group sparsity identification by HSPG.

**Image Classification:** We now consider the popular Deep Convolutional Neural Networks (DC-NNs) for image classification tasks. Specifically, we select several popular and benchmark DCNN architectures, *i.e.*, VGG16 (Simonyan & Zisserman, 2014), ResNet18 (He et al., 2016) and MobileNetV1 (Howard et al., 2017) on two benchmark datasets CIFAR10 (Krizhevsky & Hinton, 2009) and Fashion-MNIST (Xiao et al., 2017). We conduct all experiments for 300 epochs with a mini-batch size of 128 and $\lambda$ as $10^{-3}$, since it returns competitive testing accuracy to the models trained without regularization, (see more in Appendix D.3). The step size $\alpha_k$ is initialized as 0.1, and decayed by a factor 0.1 periodically. We set each kernel in the convolution layers as a group variable.

In these experiments, we proceed a test on the objective value stationarity similarly to (Zhang et al., 2020, Section 2.1) and switch to Half-Space Step roughly on 150 epochs with $N_{\mathcal{P}}$ as $150N/|\mathcal{B}|$. The control parameter $\epsilon$ in the half-space projection (9) controls the aggressiveness level of group sparsity promotion, which is first set as 0, then fined tuned to be around 0.02 to favor the sparsity level whereas does not hurt the target objective $\Psi$; the detailed procedure is in Appendix D.3. We exclude RDA because of no acceptable results attained during our tests with the step size parameter $\gamma$ setting throughout all powers of 10 from $10^{-3}$ to $10^3$, and skip Prox-Spider and SAGA since Prox-SVRG has been a superb representative to the proximal incremental gradient methods.

Table 1 demonstrates the effectiveness and superiority of HSPG, where we mark the best values as bold, and the group sparsity ratio is defined as the percentage of zero groups. In particular, *(i)* HSPG computes remarkably higher group sparsity than other methods on all tests under both $\epsilon = 0$ and fine tuned $\epsilon$, of which the solutions are typically multiple times sparser in the manner of group than those of Prox-SG, while Prox-SVRG performs not comparably since the variance reduction techniques may

Table 1: Final $\Psi$/group sparsity ratio/testing accuracy for tested algorithms on non-convex problems.

| Backbone | Dataset | Prox-SG | Prox-SVRG | HSPG | |
|---|---|---|---|---|---|
| | | | | $\epsilon$ as 0 | fine tuned $\epsilon$ |
| VGG16 | CIFAR10 | **0.59** / 53.95% / 90.57% | 0.82 / 14.73% / 89.42% | **0.59** / 74.60% / **91.10%** | **0.59** / **75.61%** / 90.92% |
| | Fashion-MNIST | 0.54 / 15.63% / **92.99%** | 2.66 / 0.45% / 92.69% | 0.54 / 22.18% / 92.98% | **0.53** / **60.77%** / 92.87% |
| ResNet18 | CIFAR10 | **0.31** / 19.50% / 94.09% | 0.36 / 2.79% / 94.17% | **0.31** / 41.58% / 94.39% | **0.31** / **62.97%** / **94.53%** |
| | Fashion-MNIST | 0.14 / 0.00% / 94.82% | 0.19 / 0.00% / 94.64% | **0.13** / 6.60% / **94.93%** | **0.13** / **63.93%** / 94.86% |
| MobileNetV1 | CIFAR10 | **0.40** / 57.81% / 91.60% | 0.65 / 32.22% / 90.08% | **0.40** / 65.04% / **91.86%** | 0.41 / **71.66%** / 91.54% |
| | Fashion-MNIST | 0.22 / 65.80% / 94.36% | 0.48 / 38.76% / 93.95% | 0.23 / 74.52% / 94.43% | 0.24 / **83.71%** / **94.44%** |

(a) Objective $\Psi$     (b) Group Sparsity Ratio     (c) Testing Accuracy     (d) HSPG VS Truncation

Figure 2: On ResNet18 with CIFAR10, (a)-(c): Evolution of $\Psi$, group sparsity ratio and testing accuracy, (d): HSPG versus Prox-SG* and Prox-SVRG* (Prox-SG and Prox-SVRG with simple truncation mechanism).

not work as desired for deep learning applications (Defazio & Bottou, 2019); *(ii)* HSPG performs competitively with respect to the final objective values $\Psi$ and $f$ (see $f$ in Appendix). In addition, all the methods reach a comparable generalization performance on unseen test data. On the other hand, sparse regularization methods may yield solutions with entries that are not exactly zero but are very small. Sometimes all entries below certain threshold ($\mathcal{T}$) are set to zero (Jenatton et al., 2010; Halabi et al., 2018). However, such simple truncation mechanism is heuristic-rule based, hence may hurt convergence and accuracy. To illustrate this, we set the groups of the solutions of Prox-SG and Prox-SVRG to zero if the magnitudes of the group variables are less than some $\mathcal{T}$, and denote the corresponding solutions as Prox-SG* and Prox-SVRG*. As shown in Figure 2d(i), under the $\mathcal{T}$ with no accuracy regression, Prox-SG* and Prox-SVRG* reach higher group sparsity ratio as 60% and 32% compared to Table 1, but still significantly lower than the 70% of HSPG under $\epsilon = 0.05$ without simple truncation. Under the $\mathcal{T}$ to reach the same group sparsity ratio as HSPG, the testing accuracy of Prox-SG* and Prox-SVRG* regresses drastically to 28% and 17% in Figure 2d(ii) respectively. Remark here that although further refitting the models from Prox-SG* and Prox-SVRG* on active (non-zero) groups of weights may recover the accuracy regression, it requires additional engineering efforts and training cost, which is less attractive and convenient than HSPG (with no need to refit).

Finally, we investigate the group sparsity evolution under different $\epsilon$'s. As shown in Figure 2b, HSPG produces the highest group-sparse solutions compared with other methods. Notably, at the early $N_{\mathcal{P}}$ iterations, HSPG performs merely the same as Prox-SG. However, after switching to Half-Space Step at the 150th epoch, HSPG outperforms all the other methods dramatically, and larger $\epsilon$ results in higher sparsity level. It is a strong evidence that our half-space based technique is much more successful than the proximal mechanism and its variants in terms of the group sparsity identification. Besides, the evolutions of $\Psi$ and testing accuracy confirm the comparability on convergence among the tested algorithms. Particularly, the objective $\Psi$ generally monotonically decreases for small $\epsilon = 0$ to 0.02, and experiences a mild pulse after switch to Half-Space Step for larger $\epsilon$, *e.g.*, 0.05, which matches Lemma 1. As a result, with the similar generalization accuracy, HSPG allows dropping entire hidden units of networks, which may further achieve automatic dimension reduction and construct smaller model architectures for efficient inference.

## 5    CONCLUSIONS AND FUTURE WORK

We proposed a new Half-Space Stochastic Projected Gradient (HSPG) method for disjoint group-sparsity induced regularized problem, which can be applied to various structured sparsity stochastic learning problem. HSPG makes use of proximal stochastic gradient method to seek a near-optimal solution estimate, followed by a novel half-space group projection to effectively exploit the group sparsity structure. In theory, we provided the convergence guarantee, and showed its better sparsity identification performance. Experiments on both convex and non-convex problems demonstrated that HSPG usually achieves solutions with competitive objective values and significantly higher group sparsity compared with state-of-the-arts stochastic solvers. Further study is needed to investigate the proper leverage of group sparsity into diverse deep learning applications, *e.g.*, help people design and understand optimal network architecture by removing redundant hidden structures.

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

## A    PROJECTION REGION

In this Appendix, we derive the projection region of HSPG, and reveal that is a superset of those of Prox-SG, Prox-SVRG and Prox-Spider under the same $\alpha_k$ and $\lambda$.

**Proposition 1.** *The Half-Space Step of HSPG yields next iterate $\boldsymbol{x}_{k+1}$ based on the trial iterate $\hat{\boldsymbol{x}}_{k+1} = \boldsymbol{x}_k - \alpha_k \nabla f_{\mathcal{B}_k}(\boldsymbol{x}_k)$ as follows for each $g \in \mathcal{I}^{\neq 0}(\boldsymbol{x}_k)$*

$$[\boldsymbol{x}_{k+1}]_g = \begin{cases} [\hat{\boldsymbol{x}}_{k+1}]_g - \alpha_k \lambda \frac{[\boldsymbol{x}_k]_g}{\|[\boldsymbol{x}_k]_g\|} & \textit{if } [\hat{\boldsymbol{x}}_{k+1}]_g^\top [\boldsymbol{x}_k]_g > (\alpha_k \lambda + \epsilon) \, \|[\boldsymbol{x}_k]_g\| \\ 0 & \textit{otherwise.} \end{cases} \tag{11}$$

*Consequently, if $\|[\hat{\boldsymbol{x}}_{k+1}]_g\| \leq \alpha_k \lambda$, then $[\boldsymbol{x}_{k+1}]_g = 0$ for any $\epsilon \geq 0$.*

*Proof.* For $g \in \mathcal{I}^{\neq 0}(\boldsymbol{x}_k) \bigcap \mathcal{I}^{\neq 0}(\boldsymbol{x}_{k+1})$, by Algorithm 3, it is equivalent to

$$\left[ \boldsymbol{x}_k - \alpha_k \nabla f_{\mathcal{B}_k}(\boldsymbol{x}_k) - \alpha_k \lambda \frac{[\boldsymbol{x}_k]_g}{\|[\boldsymbol{x}_k]_g\|} \right]_g^\top [\boldsymbol{x}_k]_g > \epsilon \, \|[\boldsymbol{x}_k]_g\|^2 ,$$
$$[\hat{\boldsymbol{x}}_{k+1}]_g^\top [\boldsymbol{x}_k]_g - \alpha_k \lambda \, \|[\boldsymbol{x}_k]_g\| > \epsilon \, \|[\boldsymbol{x}_k]_g\|^2 , \tag{12}$$
$$[\hat{\boldsymbol{x}}_{k+1}]_g^\top [\boldsymbol{x}_k]_g > (\alpha_k \lambda + \epsilon \, \|[\boldsymbol{x}_k]_g\|) \, \|[\boldsymbol{x}_k]_g\| .$$

Similarly, $g \in \mathcal{I}^{\neq 0}(\boldsymbol{x}_k) \bigcap \mathcal{I}^0(\boldsymbol{x}_{k+1})$ is equivalent to

$$
\left[ \boldsymbol{x}_k - \alpha_k \nabla f_{\mathcal{B}_k}(\boldsymbol{x}_k) - \alpha_k \lambda \frac{[\boldsymbol{x}_k]_g}{\|[\boldsymbol{x}_k]_g\|} \right]_g^\top [\boldsymbol{x}_k]_g \leq \epsilon \|[\boldsymbol{x}_k]_g\|^2 ,
$$

$$
[\hat{\boldsymbol{x}}_{k+1}]_g^\top [\boldsymbol{x}_k]_g - \alpha_k \lambda \|[\boldsymbol{x}_k]_g\| \leq \epsilon \|[\boldsymbol{x}_k]_g\|^2 , \tag{13}
$$

$$
[\hat{\boldsymbol{x}}_{k+1}]_g^\top [\boldsymbol{x}_k]_g \leq (\alpha_k \lambda + \epsilon \|[\boldsymbol{x}_k]_g\|) \|[\boldsymbol{x}_k]_g\| .
$$

If $\|[\hat{\boldsymbol{x}}_{k+1}]_g\| \leq \alpha_k \lambda$, then

$$
[\hat{\boldsymbol{x}}_{k+1}]_g^\top [\boldsymbol{x}_k]_g \leq \|[\hat{\boldsymbol{x}}_{k+1}]_g\| \|[\boldsymbol{x}_k]_g\| \leq \alpha_k \lambda \|[\boldsymbol{x}_k]_g\| . \tag{14}
$$

Hence $[\boldsymbol{x}_{k+1}]_g = 0$ holds for any $\epsilon \geq 0$ by (13), which implies that the projection region of Prox-SG and its variance reduction variants, *e.g.*, Prox-SVRG, Prox-Spider and SAGA are the subsets of HSPG's. $\qquad\square$

## B    NON-LIPSCHITZ CONTINUITY OF $\nabla \Psi$ ON $\mathbb{R}^n$

The first-derivative of $\Psi(\boldsymbol{x})$ at $\boldsymbol{x} \neq 0$ can be written as

$$
\nabla \Psi(\boldsymbol{x}) = \nabla f(\boldsymbol{x}) + \lambda \sum_{g \in \mathcal{G}} \frac{[\boldsymbol{x}]_g}{\|[\boldsymbol{x}]_g\|} \tag{15}
$$

We next show $\frac{[\boldsymbol{x}]_g}{\|[\boldsymbol{x}]_g\|}$ is not Lipschitz continuous on $\mathbb{R}^n$ if $|g| \geq 2$. Take a example for $\boldsymbol{x} \in \mathbb{R}^2$, and select $\boldsymbol{x}_1 = (t, a_1 t)^\top, \boldsymbol{x}_2 = (t, a_2 t)^\top, a_1 \neq a_2$ and $t \in \mathbb{R}$. Then suppose there exists a positive constant $L < \infty$ such that Lipschitz continuity holds as follows

$$
\left\| \frac{\boldsymbol{x}_1}{\|\boldsymbol{x}_1\|} - \frac{\boldsymbol{x}_2}{\|\boldsymbol{x}_2\|} \right\| \leq L \|\boldsymbol{x}_1 - \boldsymbol{x}_2\|
$$

$$
\left\| \frac{(1, a_1)^\top}{\sqrt{1 + a_1^2}} - \frac{(1, a_2)^\top}{\sqrt{1 + a_2^2}} \right\| \leq L |a_1 - a_2| \cdot |t| \tag{16}
$$

holds for any $t \in \mathbb{R}$, and note the left hand side is a positive constant. However, letting $t \to 0$, we have that $L \to \infty$ which contradicts the $L < \infty$. Therefore, $\frac{[\boldsymbol{x}]_g}{\|[\boldsymbol{x}]_g\|}$ is not Lipschitz continuous on $\mathbb{R}^2$, specifically the region surrounding the origin point.

Although $[\nabla \Psi(\boldsymbol{x})]_{\mathcal{I}^{\neq 0}(\boldsymbol{x})}$ is not Lipscthiz continuous on $\mathbb{R}^n$, the Lipschitz continuity still holds on by excluding a fixed size $\ell_2$-ball centered at the origin for the group of non-zero variables $\mathcal{I}^{\neq 0}(\boldsymbol{x})$ from $\mathbb{R}^n$. For our paper, we define the region where Lipscthiz continuity of $[\nabla \Psi(\boldsymbol{x})]_{\mathcal{I}^{\neq 0}(\boldsymbol{x})}$ still holds as

$$
\mathcal{X} = \{\boldsymbol{x} : \|[\boldsymbol{x}]_g\| \geq \delta_1 \text{ for each } g \in \mathcal{I}^{\neq 0}(\boldsymbol{x}), \text{ and } [\boldsymbol{x}]_g = 0 \text{ for each } g \in \mathcal{I}^0(\boldsymbol{x})\}. \tag{17}
$$

## C    CONVERGENCE ANALYSIS PROOF

Denote the sets of groups which are projected or not onto zero as

$$
\hat{\mathcal{G}}_k := \mathcal{I}^{\neq 0}(\boldsymbol{x}_k) \bigcap \mathcal{I}^0(\boldsymbol{x}_{k+1}), \text{ and} \tag{18}
$$

$$
\tilde{\mathcal{G}}_k := \mathcal{I}^{\neq 0}(\boldsymbol{x}_k) \bigcap \mathcal{I}^{\neq 0}(\boldsymbol{x}_{k+1}). \tag{19}
$$

Denote $\mathcal{X} := \{\boldsymbol{x} : \|[\boldsymbol{x}]_g\| \geq \delta_1 \text{ for each } g \in \mathcal{G}\}$ where the Lipschitz continuity of $\nabla \Psi_{\mathcal{B}}(\boldsymbol{x})$ still holds by excluding a $\ell_2$-ball centered at the origin with radius $\delta_1$ from $\mathbb{R}^n$. Let $M$ denote one upper bound of $\|\partial \Psi\|$ and $\|\boldsymbol{\xi}\|$.

Additionally, establishing some convergence results require the below constants to measure the least and largest magnitude of non-zero group variables in $\boldsymbol{x}^*$,

$$
0 < \delta_1 := \frac{1}{2} \min_{g \in \mathcal{I}^{\neq 0}(\boldsymbol{x}^*)} \|[\boldsymbol{x}^*]_g\|, \text{ and} \tag{20}
$$

$$
0 < \delta_2 := \frac{1}{2} \max_{g \in \mathcal{I}^{\neq 0}(\boldsymbol{x}^*)} \|[\boldsymbol{x}^*]_g\|. \tag{21}
$$

and a subsequent results of strict complementary assumption on any $\mathcal{B}$ uniformly,

$$0 < \delta_3 := \frac{1}{2} \min_{g \in \mathcal{I}^0(\boldsymbol{x}^*)} (\lambda - \|[\nabla f_{\mathcal{B}}(\boldsymbol{x}^*)]_g\|) \tag{22}$$

And denote the following frequently used constant $R$ describing the size of neighbor around $\boldsymbol{x}^*$.

$$R := \min \left\{ \frac{-(\delta_1 + 2\epsilon\delta_2) + \sqrt{(\delta_1 + 2\epsilon\delta_2)^2 - 4\epsilon^2\delta_2 + 4\epsilon\delta_1^2}}{\epsilon}, \delta_1 \right\} > 0. \tag{23}$$

**Remark:** (23) is well defined as $0 < \epsilon < \frac{\delta_1^2}{\delta_2}$, and degenerated to $\delta_1$ as $\epsilon = 0$.

## C.1 SUFFICIENT DECREASE OF PROX-SG STEP AND HALF-SPACE STEP

Our convergence analysis relies on the following sufficient decrease properties of Half-Space Step and Prox-SG Step.

**Sufficient Decrease of Half-Space Step:**   We prove the Lemma 1 as the below.

**Proof of Lemma 1:**

It follows Algorithm 3 and the definition of $\tilde{\mathcal{G}}_k$ and $\hat{\mathcal{G}}_k$ as (19) and (18) that $\boldsymbol{x}_{k+1} = \boldsymbol{x}_k + \alpha_k \boldsymbol{d}_k$ where $\boldsymbol{d}_k$ is

$$[\boldsymbol{d}_k]_g = \begin{cases} -[\partial \Psi_{\mathcal{B}_k}(\boldsymbol{x}_k)]_g & \text{if } g \in \tilde{\mathcal{G}}_k = \mathcal{I}^{\neq 0}(\boldsymbol{x}_k) \bigcap \mathcal{I}^{\neq 0}(\boldsymbol{x}_{k+1}), \\ -[\boldsymbol{x}_k]_g/\alpha_k & \text{if } g \in \hat{\mathcal{G}}_k = \mathcal{I}^{\neq 0}(\boldsymbol{x}_k) \bigcap \mathcal{I}^0(\boldsymbol{x}_{k+1}), \\ 0 & \text{otherwise.} \end{cases} \tag{24}$$

We also notice that for any $g \in \hat{\mathcal{G}}_k$, the following holds

$$\begin{aligned} [\boldsymbol{x}_k - \alpha_k \partial \Psi_{\mathcal{B}_k}(\boldsymbol{x}_k)]_g^\top [\boldsymbol{x}_k]_g &< \epsilon \|[\boldsymbol{x}_k]_g\|^2, \\ (1 - \epsilon) \|[\boldsymbol{x}_k]_g\|^2 &< \alpha_k [\partial \Psi_{\mathcal{B}_k}(\boldsymbol{x}_k)]_g^\top [\boldsymbol{x}_k]_g. \end{aligned} \tag{25}$$

For simplicity, let $\mathcal{I}_k^{\neq 0} := \mathcal{I}^{\neq 0}(\boldsymbol{x}_k)$. Since $[\boldsymbol{d}_k]_g = \boldsymbol{0}$ for any $g \in \mathcal{I}^0(\boldsymbol{x}_k)$, then by (24) and (25), we have

$$\begin{aligned} \boldsymbol{d}_k^\top \partial \Psi_{\mathcal{B}_k}(\boldsymbol{x}_k) &= [\boldsymbol{d}_k]_{\mathcal{I}_k^{\neq 0}}^\top [\partial \Psi_{\mathcal{B}_k}(\boldsymbol{x}_k)]_{\mathcal{I}_k^{\neq 0}} \\ &= -\sum_{g \in \tilde{\mathcal{G}}_k} \|[\partial \Psi_{\mathcal{B}_k}(\boldsymbol{x}_k)]_g\|^2 - \sum_{g \in \hat{\mathcal{G}}_k} \frac{1}{\alpha_k} [\boldsymbol{x}_k]_g^\top [\partial \Psi_{\mathcal{B}_k}(\boldsymbol{x}_k)]_g \\ &\leq -\sum_{g \in \tilde{\mathcal{G}}_k} \|[\partial \Psi_{\mathcal{B}_k}(\boldsymbol{x}_k)]_g\|^2 - \sum_{g \in \hat{\mathcal{G}}_k} \frac{1}{\alpha_k^2} (1 - \epsilon) \|[\boldsymbol{x}_k]_g\|^2 < 0, \end{aligned} \tag{26}$$

holds for any $\epsilon \in [0, 1)$, which implies that $\boldsymbol{d}_k$ is a descent direction for $\Psi_{\mathcal{B}_k}(\boldsymbol{x}_k)$.

Now, we start to prove the suffcient decrease of Half-Space Step. By the descent lemma, $\boldsymbol{x}_k \in \mathcal{X}$ and the Lipschitz continuity of $[\partial \Psi_{\mathcal{B}_k}]_{\mathcal{I}_k^{\neq 0}}$ on $\mathcal{X}$, we have that

$$\Psi_{\mathcal{B}_k}(\boldsymbol{x}_k + \alpha_k \boldsymbol{d}_k) \leq \Psi_{\mathcal{B}_k}(\boldsymbol{x}_k) + \alpha_k [\partial \Psi_{\mathcal{B}_k}(\boldsymbol{x}_k)]_{\mathcal{I}_k^{\neq 0}}^\top [\boldsymbol{d}_k]_{\mathcal{I}_k^{\neq 0}} + \frac{L}{2} \alpha_k^2 \left\| [\boldsymbol{d}_k]_{\mathcal{I}_k^{\neq 0}} \right\|^2. \tag{27}$$

Then it follows (24) that (27) can be rewritten as follows

$$\begin{aligned} &\Psi_{\mathcal{B}_k}(\boldsymbol{x}_k + \alpha_k \boldsymbol{d}_k) \\ \leq &\Psi_{\mathcal{B}_k}(\boldsymbol{x}_k) + \alpha_k [\partial \Psi_{\mathcal{B}_k}(\boldsymbol{x}_k)]_{\mathcal{I}_k^{\neq 0}}^\top [\boldsymbol{d}_k]_{\mathcal{I}_k^{\neq 0}} + \frac{L}{2} \alpha_k^2 \left\| [\boldsymbol{d}_k]_{\mathcal{I}_k^{\neq 0}} \right\|^2 \\ = &\Psi_{\mathcal{B}_k}(\boldsymbol{x}_k) - \sum_{g \in \tilde{\mathcal{G}}_k} \|[\partial \Psi_{\mathcal{B}_k}(\boldsymbol{x}_k)]_g\|^2 \left( \alpha_k - \frac{L}{2} \alpha_k^2 \right) - \sum_{g \in \hat{\mathcal{G}}_k} \left\{ [\partial \Psi_{\mathcal{B}_k}(\boldsymbol{x}_k)]_g^\top [\boldsymbol{x}_k]_g - \frac{L}{2} \|[\boldsymbol{x}_k]_g\|^2 \right\} \end{aligned}$$

$$\tag{28}$$

Consequently, combining with $\epsilon \in [0, 1)$ and (25), (28) can be further shown as

$$\Psi_{\mathcal{B}_k}(\boldsymbol{x}_{k+1}) \leq \Psi_{\mathcal{B}_k}(\boldsymbol{x}_k) - \left(\alpha_k - \frac{\alpha_k^2 L}{2}\right) \sum_{g \in \tilde{\mathcal{G}}_k} \|[\partial \Psi_{\mathcal{B}_k}(\boldsymbol{x}_k)]_g\|^2 - \left(\frac{1-\epsilon}{\alpha_k} - \frac{L}{2}\right) \sum_{g \in \hat{\mathcal{G}}_k} \|[\boldsymbol{x}_k]_g\|^2,$$
(29)

which completes the proof.

**Sufficient Decrease of Prox-SG Step:** The second lemma is well known for proximal operator under our notations. We include this proof for completeness.

**Lemma 2.** *Line 3 of Algorithm 2 yields that* $\boldsymbol{x}_{k+1} = \boldsymbol{x}_k - \alpha_k \boldsymbol{\xi}_{\alpha_k, \mathcal{B}_k}(\boldsymbol{x}_k)$, *where*

$$\boldsymbol{\xi}_{\alpha_k, \mathcal{B}_k}(\boldsymbol{x}_k) \in \nabla f_{\mathcal{B}_k}(\boldsymbol{x}_k) + \lambda \partial \Omega(\boldsymbol{x}_{k+1}).$$
(30)

*And the objective value* $\Psi_{\mathcal{B}_k}$ *satisfies*

$$\Psi_{\mathcal{B}_k}(\boldsymbol{x}_{k+1}) \leq \Psi_{\mathcal{B}_k}(\boldsymbol{x}_k) - \left(\alpha_k - \frac{\alpha_k^2 L}{2}\right) \|\boldsymbol{\xi}_{\alpha_k, \mathcal{B}_k}(\boldsymbol{x}_k)\|^2.$$
(31)

*Proof.* It follows from the line (3) in Algorithm 2 and the definitions of proximal operator that

$$\boldsymbol{x}_{k+1} = \arg\min_{\boldsymbol{x} \in \mathbb{R}^n} \frac{1}{2\alpha_k} \|\boldsymbol{x} - (\boldsymbol{x}_k - \alpha_k \nabla f_{\mathcal{B}_k}(\boldsymbol{x}_k))\|^2 + \lambda \Omega(\boldsymbol{x})$$

$$= \arg\min_{\boldsymbol{x} \in \mathbb{R}^n} \nabla f_{\mathcal{B}_k}(\boldsymbol{x}_k)^\top (\boldsymbol{x} - \boldsymbol{x}_k) + \lambda \Omega(\boldsymbol{x}) + \frac{1}{2\alpha_k} \|\boldsymbol{x} - \boldsymbol{x}_k\|^2$$
(32)

By the optimal condition, we have

$$\boldsymbol{0} \in \frac{1}{\alpha_k}(\boldsymbol{x}_{k+1} - \boldsymbol{x}_k) + \nabla f_{\mathcal{B}_k}(\boldsymbol{x}_k) + \lambda \partial \Omega(\boldsymbol{x}_{k+1}).$$
(33)

Since $\boldsymbol{x}_{k+1} = \boldsymbol{x}_k - \alpha_k \boldsymbol{\xi}_{\alpha_k, \mathcal{B}_k}(\boldsymbol{x}_k)$, we have

$$\boldsymbol{0} \in -\boldsymbol{\xi}_{\alpha_k, \mathcal{B}_k}(\boldsymbol{x}_k) + \nabla f_{\mathcal{B}_k}(\boldsymbol{x}_k) + \lambda \partial \Omega(\boldsymbol{x}_{k+1}),$$
(34)

which implies that

$$\boldsymbol{\xi}_{\alpha_k, \mathcal{B}_k}(\boldsymbol{x}_k) \in \nabla f_{\mathcal{B}_k}(\boldsymbol{x}_k) + \lambda \partial \Omega(\boldsymbol{x}_{k+1}).$$
(35)

And thus there exists some $\boldsymbol{v} \in \partial \Omega(\boldsymbol{x}_{k+1})$ such that

$$\boldsymbol{\xi}_{\alpha_k, \mathcal{B}_k}(\boldsymbol{x}_k) = \nabla f_{\mathcal{B}_k}(\boldsymbol{x}_k) + \lambda \boldsymbol{v}.$$
(36)

By Lipschitz continuity of $\nabla f_{\mathcal{B}_k}$ and convexity of $\Omega(\cdot)$, we have

$$f_{\mathcal{B}_k}(\boldsymbol{x}_{k+1}) = f_{\mathcal{B}_k}(\boldsymbol{x}_k - \alpha_k \boldsymbol{\xi}_{\alpha_k, \mathcal{B}_k}(\boldsymbol{x}_k))$$

$$\leq f_{\mathcal{B}_k}(\boldsymbol{x}_k) - \alpha_k \nabla f_{\mathcal{B}_k}(\boldsymbol{x}_k)^\top \boldsymbol{\xi}_{\alpha_k, \mathcal{B}_k}(\boldsymbol{x}_k) + \frac{\alpha_k^2 L}{2} \|\boldsymbol{\xi}_{\alpha_k, \mathcal{B}_k}(\boldsymbol{x}_k)\|^2$$
(37)

and

$$\lambda \Omega(\boldsymbol{x}_{k+1}) = \lambda \Omega(\boldsymbol{x}_k - \alpha_k \boldsymbol{\xi}_{\alpha_k, \mathcal{B}_k}(\boldsymbol{x}_k))$$

$$\leq \lambda \Omega(\boldsymbol{x}_k) + \lambda \boldsymbol{v}^\top (\boldsymbol{x}_k - \alpha_k \boldsymbol{\xi}_{\alpha_k, \mathcal{B}_k}(\boldsymbol{x}_k) - \boldsymbol{x}_k)$$
(38)

$$= \lambda \Omega(\boldsymbol{x}_k) - \alpha_k \lambda \boldsymbol{v}^\top \boldsymbol{\xi}_{\alpha_k, \mathcal{B}_k}(\boldsymbol{x}_k).$$

Hence, by (36), (37) and (38), the objective $\Psi_{\mathcal{B}_k}(\boldsymbol{x}_{k+1})$ satisfies

$$\Psi_{\mathcal{B}_k}(\boldsymbol{x}_{k+1}) = f_{\mathcal{B}_k}(\boldsymbol{x}_{k+1}) + \lambda \Omega(\boldsymbol{x}_{k+1})$$

$$\leq f_{\mathcal{B}_k}(\boldsymbol{x}_k) - \alpha_k \nabla f_{\mathcal{B}_k}(\boldsymbol{x}_k)^\top \boldsymbol{\xi}_{\alpha_k, \mathcal{B}_k}(\boldsymbol{x}_k) + \frac{\alpha_k^2 L}{2} \|\boldsymbol{\xi}_{\alpha_k, \mathcal{B}_k}(\boldsymbol{x}_k)\|^2 + \lambda \Omega(\boldsymbol{x}_k) - \alpha_k \lambda \boldsymbol{v}^\top \boldsymbol{\xi}_{\alpha_k, \mathcal{B}_k}(\boldsymbol{x}_k)$$

$$= \Psi_{\mathcal{B}_k}(\boldsymbol{x}_k) - \alpha_k (\nabla f_{\mathcal{B}_k}(\boldsymbol{x}_k) + \lambda \boldsymbol{v})^\top \boldsymbol{\xi}_{\alpha_k, \mathcal{B}_k}(\boldsymbol{x}_k) + \frac{\alpha_k^2 L}{2} \|\boldsymbol{\xi}_{\alpha_k, \mathcal{B}_k}(\boldsymbol{x}_k)\|^2$$

$$= \Psi_{\mathcal{B}_k}(\boldsymbol{x}_k) - \left(\alpha_k - \frac{\alpha_k^2 L}{2}\right) \|\boldsymbol{\xi}_{\alpha_k, \mathcal{B}_k}(\boldsymbol{x}_k)\|^2,$$

which completes the proof. $\square$

According to Lemma 1 and Lemma 2, the objective value on a mini-batch tends to achieve a sufficient decrease in both Prox-SG Step and Half-Space Step given $\alpha_k$ is small enough. By taking the expectation on both sides, we obtain the following result characterizing the sufficient decrease from $\Psi(\boldsymbol{x}_k)$ to $\mathbb{E}\left[\Psi(\boldsymbol{x}_{k+1})\right]$.

**Corollary 1.** *For iteration $k$, we have*

*(i) if $k$th iteration conducts Prox-SG Step, then*

$$\mathbb{E}\left[\Psi(\boldsymbol{x}_{k+1})\right] \leq \Psi(\boldsymbol{x}_k) - \left(\alpha_k - \frac{\alpha_k^2 L}{2}\right)\mathbb{E}\left[\left\|\boldsymbol{\xi}_{\alpha_k, \mathcal{B}_k}(\boldsymbol{x}_k)\right\|^2\right]. \tag{39}$$

*(ii) if $k$th iteration conducts Half-Space Step, $\boldsymbol{x}_k \in \mathcal{X}$, then*

$$\mathbb{E}\left[\Psi(\boldsymbol{x}_{k+1})\right] \leq \Psi(\boldsymbol{x}_k) - \sum_{g \in \tilde{\mathcal{G}}_k}\left(\alpha_k - \frac{\alpha_k^2 L}{2}\right)\mathbb{E}\left[\left\|\partial\Psi_{\mathcal{B}_k}(\boldsymbol{x}_k)\right\|^2\right] - \left(\frac{1-\epsilon}{\alpha_k} - \frac{L}{2}\right)\sum_{g \in \hat{\mathcal{G}}_k}\left\|[\boldsymbol{x}_k]_g\right\|^2. \tag{40}$$

Corollary 1 shows that the bound of $\Psi$ depends on step size $\alpha_k$ and norm of search direction. It further indicates that both Half-Space Step and Prox-SG Step can make some progress to optimality with proper selection of $\alpha_k$.

### C.2 PROOF OF THEOREM 1

Toward that end, we first show that if the optimal distance from $\boldsymbol{x}_k$ to the local minimizer $\boldsymbol{x}^*$ is sufficiently small, then HSPG already covers the supports of $\boldsymbol{x}^*$, *i.e.*, $\mathcal{I}^{\neq 0}(\boldsymbol{x}^*) \subseteq \mathcal{I}^{\neq 0}(\boldsymbol{x}_k)$.

**Lemma 3.** *If $\|\boldsymbol{x}_k - \boldsymbol{x}^*\| \leq R$, then $\mathcal{I}^{\neq 0}(\boldsymbol{x}^*) \subseteq \mathcal{I}^{\neq 0}(\boldsymbol{x}_k)$.*

*Proof.* For any $g \in I^{\neq 0}(\boldsymbol{x}^*)$, by the assumption of this lemma and the definition of $R$ as (23) and $\delta_1$ as (20), we have that

$$\begin{aligned}\left\|[\boldsymbol{x}^*]_g\right\| - \left\|[\boldsymbol{x}_k]_g\right\| &\leq \left\|[\boldsymbol{x}_k - \boldsymbol{x}^*]_g\right\| \leq \|\boldsymbol{x}_k - \boldsymbol{x}^*\| \leq R \leq \delta_1 \\ \left\|[\boldsymbol{x}_k]_g\right\| &\geq \left\|[\boldsymbol{x}^*]_g\right\| - \delta_1 \geq 2\delta_1 - \delta_1 = \delta_1 > 0\end{aligned} \tag{41}$$

Hence $\left\|[\boldsymbol{x}_k]_g\right\| \neq 0$, *i.e.*, $g \in \mathcal{I}^{\neq 0}(\boldsymbol{x}_k)$. Therefore, $\mathcal{I}^{\neq 0}(\boldsymbol{x}^*) \subseteq \mathcal{I}^{\neq 0}(\boldsymbol{x}_k)$. $\qquad\square$

The next lemma shows that if the distance between current iterate $\boldsymbol{x}_k$ and $\boldsymbol{x}^*$, *i.e.*, $\|\boldsymbol{x}_k - \boldsymbol{x}^*\|$ is sufficiently small, then $\boldsymbol{x}^*$ inhabits the reduced space $\mathcal{S}_k := \mathcal{S}(\boldsymbol{x}_k)$.

**Lemma 4.** *Under Assumption 1, if $0 \leq \epsilon < \frac{\delta_1^2}{\delta_2}$, $\|\boldsymbol{x}_k - \boldsymbol{x}^*\| \leq R$, then for each $g \in \mathcal{I}^{\neq 0}(\boldsymbol{x}^*)$,*

$$[\boldsymbol{x}_k]_g^\top[\boldsymbol{x}^*]_g \geq \epsilon\left\|[\boldsymbol{x}_k]_g\right\|^2 \tag{42}$$

*Consequently, it implies $\boldsymbol{x}^* \in \mathcal{S}_k$ by the definition as (7).*

*Proof.* It follows the assumption of this lemma and the definition of $R$ in (23), $\delta_1$ and $\delta_2$ in (23), (20) and (21) that for any $g \in \mathcal{I}^{\neq 0}(\boldsymbol{x}^*)$,

$$\left\|[\boldsymbol{x}_k]_g\right\| \leq \left\|[\boldsymbol{x}^*]_g\right\| + R \leq 2\delta_2 + R, \tag{43}$$

and the $\left[-(\delta_1 + 2\epsilon\delta_2) + \sqrt{(\delta_1 + 2\epsilon\delta_2)^2 - 4\epsilon^2\delta_2 + 4\epsilon\delta_1^2}\right]/\epsilon$ in (23) is actually the solution of $\epsilon z^2 + (4\epsilon\delta_2 + 2\delta_1)z + 4\epsilon\delta_2^2 - 4\delta_1^2 = 0$ regarding $z \in \mathbb{R}^+$. Then we have that

$$\begin{aligned}[\boldsymbol{x}_k]_g^\top[\boldsymbol{x}^*]_g &= [\boldsymbol{x}_k - \boldsymbol{x}^* + \boldsymbol{x}^*]_g^\top[\boldsymbol{x}^*]_g \\ &= [\boldsymbol{x}_k - \boldsymbol{x}^*]_g^\top[\boldsymbol{x}^*]_g + \left\|[\boldsymbol{x}^*]_g\right\|^2 \\ &\geq \left\|[\boldsymbol{x}^*]_g\right\|^2 - \left\|[\boldsymbol{x}_k - \boldsymbol{x}^*]_g\right\|\left\|[\boldsymbol{x}^*]_g\right\| \\ &= \left\|[\boldsymbol{x}^*]_g\right\|\left(\left\|[\boldsymbol{x}^*]_g\right\| - \left\|[\boldsymbol{x}_k - \boldsymbol{x}^*]_g\right\|\right) \\ &\geq 2\delta_1(2\delta_1 - R) \geq \epsilon(2\delta_2 + R)^2 \\ &\geq \epsilon\left\|[\boldsymbol{x}_k]_g\right\|^2\end{aligned} \tag{44}$$

holds for any $g \in \mathcal{I}^{\neq 0}(\boldsymbol{x}^*)$, where the second last inequality holds because that $2\delta_1(2\delta_1 - R) = \epsilon(2\delta_2 + R)^2$ as $R = \left[ -(\delta_1 + 2\epsilon\delta_2) + \sqrt{(\delta_1 + 2\epsilon\delta_2)^2 - 4\epsilon^2\delta_2 + 4\epsilon\delta_1^2} \right]/\epsilon$. Now combing with the definition of $\mathcal{S}_k$ as (7), we have $\boldsymbol{x}^*$ inhabits $\mathcal{S}_k$, which completes the proof. $\square$

Furthermore, if $\|\boldsymbol{x}_k - \boldsymbol{x}^*\|$ is small enough and the step size is selected properly, every recovery of group sparsity by Half-Space Step can be guaranteed as successful as stated in the following lemma.

**Lemma 5.** *Suppose* $k \geq N_{\mathcal{P}}$, $\|\boldsymbol{x}_k - \boldsymbol{x}^*\| \leq R$, $0 \leq \epsilon < \frac{2\delta_1 - R}{2\delta_2 + R}$ *and* $0 < \alpha_k \leq \frac{2\delta_1 - R - \epsilon(2\delta_2 + R)}{M}$, *then for any* $g \in \hat{\mathcal{G}}_k = \mathcal{I}^{\neq 0}(\boldsymbol{x}_k) \bigcap \mathcal{I}^0(\boldsymbol{x}_{k+1})$, $g$ *must be in* $\mathcal{I}^0(\boldsymbol{x}^*)$, *i.e.*, $g \in \mathcal{I}^0(\boldsymbol{x}^*)$.

*Proof.* To prove it by contradiction, suppose there exists some $g \in \hat{\mathcal{G}}_k$ such that $g \in \mathcal{I}^{\neq 0}(\boldsymbol{x}^*)$. Since $g \in \hat{\mathcal{G}}_k = \mathcal{I}^{\neq 0}(\boldsymbol{x}_k) \bigcap \mathcal{I}^0(\boldsymbol{x}_{k+1})$, then the group projection (9) is trigerred at $g$ such that

$$
\begin{aligned}
[\tilde{\boldsymbol{x}}_{k+1}]_g^\top [\boldsymbol{x}_k]_g &= [\boldsymbol{x}_k - \alpha \nabla \Psi_{\mathcal{B}_k}(\boldsymbol{x}_k)]_g^\top [\boldsymbol{x}_k]_g \\
&= \|[\boldsymbol{x}_k]_g\|^2 - \alpha_k [\nabla \Psi_{\mathcal{B}_k}(\boldsymbol{x}_k)]_g^\top [\boldsymbol{x}_k]_g < \epsilon \|[\boldsymbol{x}_k]_g\|^2 .
\end{aligned}
\tag{45}
$$

On the other hand, it follows the assumption of this lemma and $g \in \mathcal{I}^{\neq 0}(\boldsymbol{x}^*)$ that

$$
\|[\boldsymbol{x}_k - \boldsymbol{x}^*]_g\| \leq \|\boldsymbol{x}_k - \boldsymbol{x}^*\| \leq R
\tag{46}
$$

Combining the definition of $\delta_1$ as (20) and $\delta_2$ as (21), we have that

$$
\begin{aligned}
\|[\boldsymbol{x}_k]_g\| &\geq \|[\boldsymbol{x}^*]_g\| - R \geq 2\delta_1 - R \\
\|[\boldsymbol{x}_k]_g\| &\leq \|[\boldsymbol{x}^*]_g\| + R \leq 2\delta_2 + R
\end{aligned}
\tag{47}
$$

It then follows $0 < \alpha_k \leq \frac{2\delta_1 - R - \epsilon(2\delta_2 + R)}{M}$, where note $2\delta_1 - R - \epsilon(2\delta_2 + R) > 0$ as $R \leq \delta_1$ and $\epsilon < \frac{2\delta_1 - R}{2\delta_2 + R}$, that

$$
\begin{aligned}
[\tilde{\boldsymbol{x}}_{k+1}]_g^\top [\boldsymbol{x}_k]_g &= \|[\boldsymbol{x}_k]_g\|^2 - \alpha_k [\nabla \Psi_{\mathcal{B}_k}(\boldsymbol{x}_k)]_g^\top [\boldsymbol{x}_k]_g \\
&\geq \|[\boldsymbol{x}_k]_g\|^2 - \alpha_k \|[\nabla \Psi_{\mathcal{B}_k}(\boldsymbol{x}_k)]_g\| \|[\boldsymbol{x}_k]_g\| \\
&= \|[\boldsymbol{x}_k]_g\| (\|[\boldsymbol{x}_k]_g\| - \alpha_k \|[\nabla \Psi_{\mathcal{B}_k}(\boldsymbol{x}_k)]_g\|) \\
&\geq \|[\boldsymbol{x}_k]_g\| (\|[\boldsymbol{x}_k]_g\| - \alpha_k M) \\
&\geq \|[\boldsymbol{x}_k]_g\| [(2\delta_1 - R) - \alpha_k M] \\
&\geq \|[\boldsymbol{x}_k]_g\| \left[ (2\delta_1 - R) - \frac{2\delta_1 - R - \epsilon(2\delta_2 + R)}{M} M \right] \\
&\geq \|[\boldsymbol{x}_k]_g\| [(2\delta_1 - R) - 2\delta_1 + R + \epsilon(2\delta_2 + R)] \\
&\geq \epsilon \|[\boldsymbol{x}_k]_g\| (2\delta_2 + R) \\
&\geq \epsilon \|[\boldsymbol{x}_k]_g\|^2
\end{aligned}
\tag{48}
$$

which contradicts with (45). Hence, we conclude that any $g$ of variables projected to zero, *i.e.*, $g \in \hat{\mathcal{G}}_k = \mathcal{I}^{\neq 0}(\boldsymbol{x}_k) \bigcap \mathcal{I}^0(\boldsymbol{x}_{k+1})$ are exactly also the zeros on the optimal solution $\boldsymbol{x}^*$, *i.e.*, $g \in \mathcal{I}^0(\boldsymbol{x}^*)$. $\square$

We next present that if the iterate of Half-Space Step is close enough to the optimal solution $\boldsymbol{x}^*$, then $\boldsymbol{x}^*$ inhabits all reduced spaces constructed by the subsequent iterates of Half-Space Step with high probability. To establish this results, we require the below two lemmas. The first bounds the accumulated error because of random sampling. Here we introduce the error of gradient estimator on $\mathcal{I}^{\neq 0}(\boldsymbol{x})$ for $\Psi$ on mini-batch $\mathcal{B}$ as

$$
\boldsymbol{e}_{\mathcal{B}}(\boldsymbol{x}) := [\nabla \Psi_{\mathcal{B}}(\boldsymbol{x}) - \nabla \Psi(\boldsymbol{x})]_{\mathcal{I}^{\neq 0}(\boldsymbol{x})},
\tag{49}
$$

where by the definition of $\Omega$ in problem (1), we have $\boldsymbol{e}_{\mathcal{B}}(\boldsymbol{x})$ also equals to the error of estimation for $\nabla f$,

$$
\boldsymbol{e}_{\mathcal{B}}(\boldsymbol{x}) = [\nabla \Psi_{\mathcal{B}}(\boldsymbol{x}) - \nabla \Psi(\boldsymbol{x})]_{\mathcal{I}^{\neq 0}(\boldsymbol{x})} = [\nabla f_{\mathcal{B}}(\boldsymbol{x}) - \nabla f(\boldsymbol{x})]_{\mathcal{I}^{\neq 0}(\boldsymbol{x})}.
\tag{50}
$$

**Lemma 6.** *Given any $\theta > 1$, $K \geq N_{\mathcal{P}}$, let $k := K + t$, $t \in \mathbb{Z}^{+} \bigcup \{0\}$, then there exists $\alpha_k = \mathcal{O}(1/t)$ and $|\mathcal{B}_k| = \mathcal{O}(t)$, such that for any $y_t \in \mathbb{R}^n$,*

$$\max_{\{\boldsymbol{y}_t\}_{t=0}^{\infty} \in \mathcal{X}^{\infty}} \sum_{t=0}^{\infty} \alpha_k \| e_{\mathcal{B}_k}(\boldsymbol{y}_t) \|_2 \leq \frac{3R^2}{8(4R+1)}$$

*holds with probability at least $1 - \frac{1}{\theta^2}$.*

*Proof.* Define random variable $Y_t := \alpha_{K+t} \| e_{\mathcal{B}_{K+t}}(\boldsymbol{y}_t) \|_2$ for all $t \geq 0$. Since $\{\boldsymbol{y}_t\}_{t=0}^{\infty}$ are arbitrarily chosen, then the random variables $\{Y_t\}_{t=0}^{\infty}$ are independent. Let $Y := \sum_{t=0}^{\infty} Y_t$. Using Chebshev's inequality, we obtain

$$\mathbb{P}\left(Y \geq \mathbb{E}[Y] + \theta\sqrt{\mathrm{Var}[Y]}\right) \leq \mathbb{P}\left(|Y - \mathbb{E}[Y]| \geq \theta\sqrt{\mathrm{Var}[Y]}\right) \leq \frac{1}{\theta^2}. \tag{51}$$

And based on the Assumption 1, there exists an upper bound $\sigma^2 > 0$ for the variance of random noise $e(\boldsymbol{x})$ generated from the one-point mini-batch, *i.e.*, $\mathcal{B} = \{i\}, i = 1, \ldots, N$. Consequently, for each $t \geq 0$, we have $\mathbb{E}[Y_t] \leq \frac{\alpha_{K+t}\sigma}{\sqrt{|\mathcal{B}_{K+t}|}}$ and $\mathrm{Var}[Y_t] \leq \frac{\alpha_{K+t}^2\sigma^2}{|\mathcal{B}_{K+t}|}$, then combining with (51), we have

$$Y \leq \mathbb{E}[Y] + \theta\sqrt{\mathrm{Var}[Y]} \tag{52}$$

$$\leq \sum_{t=0}^{\infty} \frac{\alpha_{K+t}\sigma}{\sqrt{|\mathcal{B}_{k+t}|}} + \theta \cdot \sum_{t=0}^{\infty} \frac{\alpha_{K+t}^2\sigma^2}{|\mathcal{B}_{K+t}|} \tag{53}$$

$$\leq \sum_{t=0}^{\infty} \frac{\alpha_{K+t}\sigma}{\sqrt{|\mathcal{B}_{k+t}|}} + \theta \cdot \sum_{t=0}^{\infty} \frac{\alpha_{K+t}\sigma}{\sqrt{|\mathcal{B}_{K+t}|}} = (1+\theta)\sum_{t=0}^{\infty} \frac{\alpha_{K+t}\sigma}{\sqrt{|\mathcal{B}_{K+t}|}} \tag{54}$$

holds with probability at least $1 - \frac{1}{\theta^2}$. Here, for the second inequality, we use the property that the equality $\mathbb{E}[\sum_{t=0}^{\infty} Y_i] = \sum_{t=0}^{\infty} \mathbb{E}[Y_i]$ holds whenever $\sum_{t=0}^{\infty} \mathbb{E}[|Y_i|]$ convergences, see Section 2.1 in Mitzenmacher (2005); and for the third inequality, we use $\frac{\alpha_{K+t}\sigma}{\sqrt{|\mathcal{B}_{K+t}|}} \leq 1$ without loss of generality as the common setting of large mini-batch size and small step size.

Given any $\theta > 1$, there exists some $\alpha_k = \mathcal{O}(1/t)$ and $|\mathcal{B}_k| = \mathcal{O}(t)$, the above series converges and satisfies that

$$(1+\theta)\sum_{t=0}^{\infty} \frac{\alpha_{K+t}\sigma}{\sqrt{|\mathcal{B}_{K+t}|}} \leq \frac{3R^2}{8(4R+1)} \tag{55}$$

holds. Notice that the above proof holds for any given sequence $\{\boldsymbol{y}_t\}_{t=0}^{\infty} \in \mathcal{X}^{\infty}$, thus

$$\max_{\{\boldsymbol{y}_t\}_{t=0}^{\infty} \in \mathcal{X}^{\infty}} \sum_{t=0}^{\infty} \alpha_k \| e_{\mathcal{B}_k}(\boldsymbol{y}_t) \|_2 \leq \frac{3R^2}{8(4R+1)}$$

holds with probability at least $1 - \frac{1}{\theta^2}$. $\qquad \square$

The second lemma draws if previous iterate of Half-Space Step falls into the neighbor of $\boldsymbol{x}^*$, then under appropriate step size and mini-batch setting, the current iterate also inhabits the neighbor with high probability.

**Lemma 7.** *Under the assumptions of Lemma 6, suppose $\|\boldsymbol{x}_K - \boldsymbol{x}^*\| \leq R/2$; for any $\ell$ satisfying $K \leq \ell < K + t$, $0 < \alpha_\ell \leq \min\{\frac{1}{L}, \frac{2\delta_1 - R - \epsilon(2\delta_2 + R)}{M}\}$, $|B_\ell| \geq N - \frac{N}{2M}$ and $\|\boldsymbol{x}_\ell - \boldsymbol{x}^*\| \leq R$ holds, then*

$$\|\boldsymbol{x}_{K+t} - \boldsymbol{x}^*\| \leq R. \tag{56}$$

*holds with probability at least $1 - \frac{1}{\theta^2}$.*

*Proof.* It follows the assumptions of this lemma, Lemma 5, (18) and (19) that for any $\ell$ satisfying $K \leq \ell < K + t$

$$\|[\boldsymbol{x}^*]_g\| = 0, \text{ for any } g \in \hat{\mathcal{G}}_\ell. \tag{57}$$

Hence we have that for $K \leq \ell < K + t$,

$$
\begin{aligned}
&\|\boldsymbol{x}_{\ell+1} - \boldsymbol{x}^*\|^2 \\
&= \sum_{g \in \tilde{\mathcal{G}}_\ell} \|[\boldsymbol{x}_\ell - \boldsymbol{x}^* - \alpha_\ell \nabla \Psi(\boldsymbol{x}_\ell) - \alpha_\ell \boldsymbol{e}_{\mathcal{B}_\ell}(\boldsymbol{x}_\ell)]_g\|^2 + \sum_{g \in \hat{\mathcal{G}}_k} \|[\boldsymbol{x}_\ell - \boldsymbol{x}^* - \boldsymbol{x}_\ell]_g\|^2 \\
&= \sum_{g \in \tilde{\mathcal{G}}_\ell} \left\{ \|[\boldsymbol{x}_\ell - \boldsymbol{x}^*]_g\|^2 - 2\alpha_\ell [\boldsymbol{x}_\ell - \boldsymbol{x}^*]_g^\top [\nabla \Psi(\boldsymbol{x}_\ell) + \boldsymbol{e}_{\mathcal{B}_\ell}(\boldsymbol{x}_\ell)]_g + \alpha_\ell^2 \|[\nabla \Psi(\boldsymbol{x}_\ell) + \boldsymbol{e}_{\mathcal{B}_\ell}(\boldsymbol{x}_\ell)]_g\|^2 \right\} + \sum_{g \in \hat{\mathcal{G}}_\ell} \|[\boldsymbol{x}^*]_g\|^2 \\
&= \sum_{g \in \tilde{\mathcal{G}}_\ell} \left\{ \|[\boldsymbol{x}_\ell - \boldsymbol{x}^*]_g\|^2 - 2\alpha_\ell [\boldsymbol{x}_\ell - \boldsymbol{x}^*]_g^\top [\nabla \Psi(\boldsymbol{x}_\ell)]_g - 2\alpha_\ell [\boldsymbol{x}_\ell - \boldsymbol{x}^*]_g^\top [\boldsymbol{e}_{\mathcal{B}_\ell}(\boldsymbol{x}_\ell)]_g + \alpha_\ell^2 \|[\nabla \Psi(\boldsymbol{x}_\ell) + \boldsymbol{e}_{\mathcal{B}_\ell}(\boldsymbol{x}_\ell)]_g\|^2 \right\} \\
&\leq \sum_{g \in \tilde{\mathcal{G}}_\ell} \|[\boldsymbol{x}_\ell - \boldsymbol{x}^*]_g\|^2 - \|[\nabla \Psi(\boldsymbol{x}_\ell)]_g\|^2 \left( 2\frac{\alpha_\ell}{L} - \alpha_\ell^2 \right) - 2\alpha_\ell [\boldsymbol{x}_\ell - \boldsymbol{x}^*]_g^\top [\boldsymbol{e}_{\mathcal{B}_\ell}(\boldsymbol{x}_\ell)]_g + \alpha_\ell^2 \|[\boldsymbol{e}_{\mathcal{B}_\ell}(\boldsymbol{x}_\ell)]_g\|^2 \\
&\quad + 2\alpha_\ell^2 [\nabla \Psi(\boldsymbol{x}_\ell)]_g^\top [\boldsymbol{e}_{\mathcal{B}_\ell}(\boldsymbol{x}_\ell)]_g \\
&\leq \sum_{g \in \tilde{\mathcal{G}}_\ell} \|[\boldsymbol{x}_\ell - \boldsymbol{x}^*]_g\|^2 - \|[\nabla \Psi(\boldsymbol{x}_\ell)]_g\|^2 \left( 2\frac{\alpha_\ell}{L} - \alpha_\ell^2 \right) + 2\alpha_\ell \|[\boldsymbol{x}_\ell - \boldsymbol{x}^*]_g\| \|[\boldsymbol{e}_{\mathcal{B}_\ell}(\boldsymbol{x}_\ell)]_g\| + \alpha_\ell^2 \|[\boldsymbol{e}_{\mathcal{B}_\ell}(\boldsymbol{x}_\ell)]_g\|^2 \\
&\quad + 2\alpha_\ell^2 \|[\nabla \Psi(\boldsymbol{x}_\ell)]_g\| \|[\boldsymbol{e}_{\mathcal{B}_\ell}(\boldsymbol{x}_\ell)]_g\| \\
&\leq \sum_{g \in \tilde{\mathcal{G}}_\ell} \|[\boldsymbol{x}_\ell - \boldsymbol{x}^*]_g\|^2 - \|[\nabla \Psi(\boldsymbol{x}_\ell)]_g\|^2 \left( 2\frac{\alpha_\ell}{L} - \alpha_\ell^2 \right) + (2\alpha_\ell + 2\alpha_\ell^2 L) \|[\boldsymbol{x}_k - \boldsymbol{x}^*]_g\| \|[\boldsymbol{e}_{\mathcal{B}_\ell}(\boldsymbol{x}_\ell)]_g\| + \alpha_\ell^2 \|[\boldsymbol{e}_{\mathcal{B}_\ell}(\boldsymbol{x}_\ell)]_g\|^2 \\
&\leq \sum_{g \in \tilde{\mathcal{G}}_\ell} \left\{ \|[\boldsymbol{x}_\ell - \boldsymbol{x}^*]_g\|^2 - \|[\nabla \Psi(\boldsymbol{x}_\ell)]_g\|^2 \left( 2\frac{\alpha_\ell}{L} - \alpha_\ell^2 \right) \right\} + (2\alpha_\ell + 2\alpha_\ell^2 L) \|\boldsymbol{x}_\ell - \boldsymbol{x}^*\| \|\boldsymbol{e}_{\mathcal{B}_\ell}(\boldsymbol{x}_\ell)\| + \alpha_\ell^2 \|\boldsymbol{e}_{\mathcal{B}_\ell}(\boldsymbol{x}_\ell)\|^2
\end{aligned}
\tag{58}
$$

On the other hand, by the definition of $\boldsymbol{e}_{\mathcal{B}}(\boldsymbol{x})$ as (49), we have that

$$
\begin{aligned}
\boldsymbol{e}_{\mathcal{B}}(\boldsymbol{x}) &= [\nabla \Psi_{\mathcal{B}}(\boldsymbol{x}) - \nabla \Psi(\boldsymbol{x})]_{\mathcal{I} \neq 0(\boldsymbol{x})} = [\nabla f_{\mathcal{B}}(\boldsymbol{x}) - \nabla f(\boldsymbol{x})]_{\mathcal{I} \neq 0(\boldsymbol{x})} \\
&= \frac{1}{|\mathcal{B}|} \sum_{j \in \mathcal{B}} [\nabla f_j(\boldsymbol{x})]_{\mathcal{I} \neq 0(\boldsymbol{x})} - \frac{1}{N} \sum_{i=1}^N [\nabla f_i(\boldsymbol{x})]_{\mathcal{I} \neq 0(\boldsymbol{x})} \\
&= \frac{1}{N} \sum_{j \in \mathcal{B}} \left[ \frac{N}{|\mathcal{B}|} [\nabla f_j(\boldsymbol{x})]_{\mathcal{I} \neq 0(\boldsymbol{x})} - [\nabla f_j(\boldsymbol{x})]_{\mathcal{I} \neq 0(\boldsymbol{x})} \right] - \frac{1}{N} \sum_{\substack{i=1 \\ i \notin \mathcal{B}}}^N [\nabla f_i(\boldsymbol{x})]_{\mathcal{I} \neq 0(\boldsymbol{x})} \\
&= \frac{1}{N} \sum_{j \in \mathcal{B}} \left[ \frac{N - |\mathcal{B}|}{|\mathcal{B}|} [\nabla f_j(\boldsymbol{x})]_{\mathcal{I} \neq 0(\boldsymbol{x})} \right] - \frac{1}{N} \sum_{\substack{i=1 \\ i \notin \mathcal{B}}}^N [\nabla f_i(\boldsymbol{x})]_{\mathcal{I} \neq 0(\boldsymbol{x})}
\end{aligned}
\tag{59}
$$

Thus taking the norm on both side of (59) and using triangle inequality results in the following:

$$
\begin{aligned}
\|\boldsymbol{e}_{\mathcal{B}}(\boldsymbol{x})\| &\leq \frac{1}{N} \sum_{j \in \mathcal{B}} \left[ \frac{N - |\mathcal{B}|}{|\mathcal{B}|} \|[\nabla f_j(\boldsymbol{x})]_{\mathcal{I} \neq 0(\boldsymbol{x})}\| \right] + \frac{1}{N} \sum_{\substack{i=1 \\ i \notin \mathcal{B}}}^N \|[\nabla f_i(\boldsymbol{x})]_{\mathcal{I} \neq 0(\boldsymbol{x})}\| \\
&\leq \frac{1}{N} \frac{N - |\mathcal{B}|}{|\mathcal{B}|} |\mathcal{B}_k| M + \frac{1}{N} (N - |\mathcal{B}|) M \leq \frac{2(N - |\mathcal{B}|)M}{N}.
\end{aligned}
\tag{60}
$$

Since $\alpha_\ell \leq 1$, and $|B_\ell| \geq N - \frac{N}{2M}$ hence $\alpha_\ell \|e_{\mathcal{B}_\ell}(x_\ell)\| \leq 1$. Then combining with $\alpha_\ell \leq 1/L$, (58) can be further simplified as

$$
\begin{aligned}
&\|x_{\ell+1} - x^*\|^2 \\
&\leq \sum_{g \in \tilde{\mathcal{G}}_\ell} \left\{ \|[x_\ell - x^*]_g\|^2 - \|[\nabla\Psi(x_\ell)]_g\|^2 \left( 2\frac{\alpha_\ell}{L} - \alpha_\ell^2 \right) \right\} + (2\alpha_\ell + 2\alpha_\ell^2 L) \|x_\ell - x^*\| \, \|e_{\mathcal{B}_\ell}(x_\ell)\| + \alpha_\ell^2 \|e_{\mathcal{B}_\ell}(x_\ell)\|^2 \\
&\leq \sum_{g \in \tilde{\mathcal{G}}_\ell} \left\{ \|[x_\ell - x^*]_g\|^2 - \frac{1}{L^2} \|[\nabla\Psi(x_\ell)]_g\|^2 \right\} + 4\alpha_\ell \|x_\ell - x^*\| \, \|e_{\mathcal{B}_\ell}(x_\ell)\| + \alpha_\ell^2 \|e_{\mathcal{B}_\ell}(x_\ell)\|^2 \\
&\leq \|x_\ell - x^*\|^2 + 4\alpha_\ell \|x_\ell - x^*\| \, \|e_{\mathcal{B}_\ell}(x_\ell)\| + \alpha_\ell \|e_{\mathcal{B}_\ell}(x_\ell)\|
\end{aligned}
\tag{61}
$$

Following from the assumption that $\|x_\ell - x^*\| \leq R$, then (61) can be further simplified as

$$
\begin{aligned}
\|x_{\ell+1} - x^*\|^2 &\leq \|x_\ell - x^*\|^2 + 4\alpha_\ell R \|e_{\mathcal{B}_\ell}(x_\ell)\| + \alpha_k \|e_{\mathcal{B}_\ell}(x_\ell)\| \\
&\leq \|x_\ell - x^*\|^2 + (4R + 1)\alpha_\ell \|e_{\mathcal{B}_\ell}(x_\ell)\|
\end{aligned}
\tag{62}
$$

Summing the the both side of (62) from $\ell = K$ to $\ell = K + t - 1$ results in

$$
\|x_{K+t} - x^*\|^2 \leq \|x_K - x^*\|^2 + (4R + 1) \sum_{\ell=K}^{K+t-1} \alpha_\ell \|e_{\mathcal{B}_\ell}(x_\ell)\|
\tag{63}
$$

It follows Lemma 6 that the followng holds with probability at least $1 - \frac{1}{\theta^2}$,

$$
\sum_{\ell=K}^{\infty} \alpha_\ell \|e_{\mathcal{B}_\ell}(x_\ell)\| \leq \frac{3R^2}{4(4R + 1)}.
\tag{64}
$$

Thus we have that

$$
\begin{aligned}
\|x_{K+t} - x^*\|^2 &\leq \|x_K - x^*\|^2 + (4R + 1) \sum_{\ell=K}^{K+t-1} \alpha_\ell \|e_{\mathcal{B}_\ell}(x_\ell)\| \\
&\leq \|x_K - x^*\|^2 + (4R + 1) \sum_{\ell=K}^{\infty} \alpha_\ell \|e_{\mathcal{B}_\ell}(x_\ell)\| \\
&\leq \frac{R^2}{4} + (4R + 1)\frac{3R^2}{4(4R + 1)} \leq \frac{R^2}{4} + \frac{3R^2}{4} \leq R^2,
\end{aligned}
\tag{65}
$$

holds with probability at least $1 - \frac{1}{\theta^2}$, which completes the proof.

$\square$

Based on the above lemmas, the Lemma 8 below shows if initial iterate of Half-Space Step locates closely enough to $x^*$, step size $\alpha_k$ polynomially decreases, and mini-batch size $\mathcal{B}_k$ polynomially increases, then $x^*$ inhabits all subsequent reduced space $\{\mathcal{S}_k\}_{k=K}^{\infty}$ constructed in Half-Space Step with high probability.

**Lemma 8.** *Suppose $\|x_K - x^*\| \leq \frac{R}{2}$, $K \geq N_{\mathcal{P}}$, $k = K + t$, $t \in \mathbb{Z}^+$, $0 < \alpha_k = \mathcal{O}(1/(\sqrt{N}t)) \leq \min\{\frac{2(1-\epsilon)}{L}, \frac{1}{L}, \frac{2\delta_1 - R - \epsilon(2\delta_2 + R)}{M}\}$ and $|\mathcal{B}_k| = \mathcal{O}(t) \geq N - \frac{N}{2M}$. Then for any constant $\tau \in (0, 1)$, $\|x_k - x^*\| \leq R$ with probability at least $1 - \tau$ for any $k \geq K$.*

*Proof.* It follows Lemma 4 and the assumption of this lemma that $x^* \in \mathcal{S}_K$. Moreover, it follows the assumptions of this lemma, Lemma 6 and 7, the definition of finite-sum $f(x)$ in (1), and the bound of error as (60) that

$$
\mathbb{P}(\{x_k\}_{k=K}^{\infty} \in \{x : \|x - x^*\| \leq R\}^{\infty}) \geq \left( 1 - \frac{1}{\theta^2} \right)^{\mathcal{O}(N-K)} \geq 1 - \tau,
\tag{66}
$$

where the last two inequalities comes from that the error vanishing to zero as $|\mathcal{B}_k|$ reaches the upper bound $N$, and $\theta$ is sufficiently large depending on $\tau$ and $\mathcal{O}(N - K)$. $\square$

**Corollary 2.** *Lemma 8 further implies $x^*$ inhabits all subsequent $\mathcal{S}_k$, i.e., $x^* \in \mathcal{S}_k$ for any $k \geq K$.*

Next, we establish that after finitely number of iterations, HSPG generates sequences that inhabits in the feasible domain $\mathcal{X}$ where Lipschitz continuity of $\Psi$ holds.

**Lemma 9.** *Suppose the assumptions of Lemma 8 hold, then after finite number of iterations, all subsequent iterates $x_k \in \mathcal{X}$ with high probability.*

*Proof.* It follows Lemma 8 that all subsequent $x_k$ satisfying $\|x_k - x^*\| \leq R$ with high probability. Combining with Lemma 3, we have that $\mathcal{I}^{\neq 0}(x^*) \subseteq \mathcal{I}^{\neq 0}(x_k)$ for all $k \geq K$ with high probability. Then for any $g \in \mathcal{I}^{\neq 0}(x_k)$, there are two possbilities, either $g \in \mathcal{I}^{\neq 0}(x^*)$ or $g \in \mathcal{I}^{0}(x^*)$. For the first case $g \in \mathcal{I}^{\neq 0}(x^*) \bigcap \mathcal{I}^{\neq 0}(x_k)$, it follows the definitions of $R$ as (23) and $\delta_1$ as (20) that

$$\|[x_k - x^*]_g\| \leq \|x_k - x^*\| \leq R \leq \delta_1$$
$$\|[x^*]_g\| - \|[x_k]_g\| \leq \delta_1 \tag{67}$$
$$\|[x_k]_g\| \geq \|[x^*]_g\| - \delta_1 \geq 2\delta_1 - \delta_1 = \delta_1$$

For any $g \in \mathcal{I}^{0}(x^*) \bigcap \mathcal{I}^{\neq 0}(x_k)$, by Algorithm 3, its norm is bounded below by

$$\delta_1 \geq \|[x_k - x^*]_g\| = \|[x_k]_g\| \geq \epsilon^t \|[x_K]_g\|, \tag{68}$$

where by the Theorem 2 will shown in Appendix C.3, if $\|[x_k]_g\| \leq \frac{2\alpha_k \delta_3}{1 - \epsilon + \alpha_k L}$, then $[x_{k+1}]_g$ equals to zero and will be fixed as zero since Algorithm 3 operates on $\mathcal{S}_k$ as (7). Note $\alpha_k = \mathcal{O}(1/t)$, following (Karimi et al., 2016, Theorem 4) and (Drusvyatskiy & Lewis, 2018, Theorem 3.2), $\mathbb{E}[\|[x_k]_g\|^2] = \mathcal{O}(1/t)$. If $\epsilon > 0$, then after finite number of iterations $\mathcal{O}(1/\epsilon^2)$, $g \in \mathcal{I}^{0}(x^*) \bigcap \mathcal{I}^{\neq 0}(x_k)$ becomes zero. If $\epsilon = 0$, note $\mathcal{B}_k = \mathcal{O}(t)$ and $f$ is finite-sum, then similar result holds by (Gower, 2018, Theorem 2.3, Theorem 3.2) ($f$ needs further strongly convexity on $\tilde{\mathcal{X}}$). Hence with high probability, after finite number of iterations, denoted by $T$, all subsequent $x_k$, $k \geq K + T$ inhabits $\mathcal{X}$. Regarding $[x_k]_{g \in \mathcal{I}^{0}(x^*) \bigcap \mathcal{I}^{\neq 0}(x_k)}$ for $K \leq k \leq K + T$, note $\epsilon^t \|[x_K]_g\|$ is also bounded below by constant $\epsilon^T \|[x_K]_g\| > 0$ given $x_K$, for simplicity, denote the Lipschitz constant of $[\nabla \Psi(x_k)]_g$ as $L$ as well. $\square$

We now prove the first main theorem of HSPG, *i.e.*, Theorem 1.

**Proof of Theorem 1:**

We know that Algorithm 1 performs an infinite sequence of iterations. It follows Corollary 1 that for any $\ell \in \mathbb{Z}^+$,

$$\mathbb{E}[\Psi(x_K)] - \mathbb{E}[\Psi(x_{\ell+1})] = \sum_{k=K}^{\ell} \{\mathbb{E}[\Psi(x_k)] - \mathbb{E}[\Psi(x_{k+1})]\}$$
$$\geq \sum_{K \leq k \leq \ell} \left(\alpha_k - \frac{\alpha_k^2 L}{2}\right) \sum_{g \in \tilde{\mathcal{G}}_k} \mathbb{E}\left[\|[\nabla \Psi(x_k)]_g\|^2\right] + \sum_{K \leq k \leq \ell} \left(\frac{1 - \epsilon}{\alpha_k} - \frac{L}{2}\right) \sum_{g \in \hat{\mathcal{G}}_k} \|[x_k]_g\|^2. \tag{69}$$

Combining the assumption that $\Psi$ is bounded below and letting $\ell \to \infty$, we obtain

$$\sum_{k \geq K} \left(\alpha_k - \frac{\alpha_k^2 L}{2}\right) \sum_{g \in \tilde{\mathcal{G}}_k} \mathbb{E}\left[\|[\nabla \Psi(x_k)]_g\|^2\right] + \sum_{k \geq K} \left(\frac{1 - \epsilon}{\alpha_k} - \frac{L}{2}\right) \sum_{g \in \hat{\mathcal{G}}_k} \|[x_k]_g\|^2 < \infty \tag{70}$$

By Algorithm 3, variables on $\mathcal{I}^{0}(x_k)$ are fixed during $k$th Half-Space Step and $n$ is finite, then the group projection appears finitely many times, consequently,

$$\sum_{k \geq K} \left(\frac{1 - \epsilon}{\alpha_k} - \frac{L}{2}\right) \sum_{g \in \hat{\mathcal{G}}_k} \|[x_k]_g\|^2 < \infty. \tag{71}$$

Thus (70) implies that

$$\sum_{k \geq K} \left(\alpha_k - \frac{\alpha_k^2 L}{2}\right) \sum_{g \in \tilde{\mathcal{G}}_k} \mathbb{E}\left[\|[\nabla \Psi(x_k)]_g\|^2\right] \tag{72}$$

$$= \sum_{k \geq K} \alpha_k \sum_{g \in \tilde{\mathcal{G}}_k} \mathbb{E}\left[\|[\nabla \Psi(x_k)]_g\|^2\right] - \sum_{k \geq K} \frac{\alpha_k^2}{L} \sum_{g \in \tilde{\mathcal{G}}_k} \mathbb{E}\left[\|[\nabla \Psi(x_k)]_g\|^2\right] < \infty \tag{73}$$

Since $\alpha_k = \mathcal{O}(1/(\sqrt{N}t))$, then $\sum_{k \geq K} \alpha_k = \infty$ and $\sum_{k \geq K} \alpha_k^2 \leq \infty$. Combining with (72) and the boundness of $\partial \Psi$, it implies

$$\sum_{k \geq K} \alpha_k \sum_{g \in \tilde{\mathcal{G}}_k} \mathbb{E}\left[\|[\nabla \Psi(\boldsymbol{x}_k)]_g\|^2\right] < \infty. \tag{74}$$

By $\sum_{k \geq K} \alpha_k = \infty$ and (74), we have that

$$\liminf_{k \geq K} \sum_{g \in \tilde{\mathcal{G}}_k} \mathbb{E}\left[\|[\nabla \Psi(\boldsymbol{x}_k)]_g\|^2\right] = 0 \tag{75}$$

then there exists a subsequence $\mathcal{K}$ such that

$$\lim_{k \in \mathcal{K}} \sum_{g \in \tilde{\mathcal{G}}_k} \mathbb{E}\left[\|[\nabla \Psi(\boldsymbol{x}_k)]_g\|^2\right] = 0 \tag{76}$$

It follows from the assumptions of this theorem and Lemma 3 to 8 and Corollay 2 that with high probability at least $1 - \tau$, for each $k \geq K$, $\boldsymbol{x}^*$ inhabits $\mathcal{S}_k$. Note as $|\mathcal{B}_k| = \mathcal{O}(t)$ linearly increases, the error of gradient estimate vanishes. Hence, (76) naturally implies that the sequence $\{\boldsymbol{x}_k\}_{k \in \mathcal{K}}$ converges to some stationary point with high probability. And we can extend $\mathcal{K}$ to $\{k : k \geq K\}$ due to the non-decreasing distance to optimal solution as shown in the Lemma 8. By the above, we conclude that

$$\mathbb{P}(\lim_{k \to \infty} \mathbb{E}\left[\|\boldsymbol{\xi}_{\alpha_k, \mathcal{B}_k}(\boldsymbol{x}_k)\|\right] = 0) \geq 1 - \tau. \tag{77}$$

### C.3 PROOF OF THEOREM 2

In this Appendix, we compare the group sparsity identification property of HSPG and Prox-SG. We first show the generic sparsity identification property of Prox-SG for any mixed $\ell_1/\ell_p$ regularization for $p \geq 1$.

**Lemma 10.** *If $\|\boldsymbol{x}_k - \boldsymbol{x}^*\|_{p'} \leq \min\{\delta_3/L, \alpha_k \delta_3\}$, where $1/p + 1/p' = 1$ ($p' = \infty$ if $p = 1$), then the Prox-SG yields that for each $g \in \mathcal{I}^0(\boldsymbol{x}^*)$, $[\boldsymbol{x}_{k+1}]_g = 0$ holds, i.e., $\mathcal{I}^0(\boldsymbol{x}^*) \subseteq \mathcal{I}^0(\boldsymbol{x}_{k+1})$.*

*Proof.* It follows from the reverse triangle inequality, basic norm inequalities, Lipschitz continuity of $\nabla f(\boldsymbol{x})$ and the assumption of this lemma that for any $g \in \mathcal{G}$,

$$\begin{aligned}
\|[\nabla f_{\mathcal{B}_k}(\boldsymbol{x}_k)]_g\|_{p'} - \|[\nabla f_{\mathcal{B}_k}(\boldsymbol{x}^*)]_g\|_{p'} &\leq \|[\nabla f_{\mathcal{B}_k}(\boldsymbol{x}_k) - \nabla f_{\mathcal{B}_k}(\boldsymbol{x}^*)]_g\|_{p'} \\
&\leq \|\nabla f_{\mathcal{B}_k}(\boldsymbol{x}_k) - \nabla f_{\mathcal{B}_k}(\boldsymbol{x}^*)\|_{p'} \\
&\leq L \|\boldsymbol{x}_k - \boldsymbol{x}^*\|_{p'} \leq L \cdot \frac{\delta_3}{L} = \delta_3.
\end{aligned} \tag{78}$$

By (78), we have that for any $g \in \mathcal{I}^0(\boldsymbol{x}^*)$,

$$\begin{aligned}
\|[\nabla f_{\mathcal{B}_k}(\boldsymbol{x}_k)]_g\|_{p'} &\leq \|[\nabla f_{\mathcal{B}_k}(\boldsymbol{x}^*)]_g\|_{p'} + \delta_3 \\
&\leq \lambda - 2\delta_3 + \delta_3 = \lambda - \delta_3
\end{aligned} \tag{79}$$

Combining (79) and the assumption of this lemma, the following holds for any $\alpha_k > 0$ that

$$\begin{aligned}
\|[\boldsymbol{x}_k - \alpha_k \nabla f_{\mathcal{B}_k}(\boldsymbol{x}_k)]_g\|_{p'} &\leq \|[\boldsymbol{x}_k]_g\|_{p'} + \|[\alpha_k \nabla f_{\mathcal{B}_k}(\boldsymbol{x}_k)]_g\|_{p'} \\
&\leq \alpha_k \delta_3 + \alpha_k(\lambda - \delta_3) = \alpha_k \lambda
\end{aligned} \tag{80}$$

which further implies that the Ecludiean projection yields that

$$\mathrm{Proj}^e_{\mathcal{B}(\|\cdot\|_{p'}, \alpha_k \lambda)}([\boldsymbol{x}_k - \alpha_k \nabla f_{\mathcal{B}_k}(\boldsymbol{x}_k)]_g) = [\boldsymbol{x}_k - \alpha_k \nabla f_{\mathcal{B}_k}(\boldsymbol{x}_k)]_g. \tag{81}$$

Combining with (81), the fact that proximal operator is the residual of identity operator subtracted by Euclidean project operator onto the dual norm ball and $[\boldsymbol{x}_k]_g = 0$ for any $g \in \mathcal{I}^0(\boldsymbol{x}^*)$ (Chen, 2018), we have that

$$\begin{aligned}
[\boldsymbol{x}_{k+1}]_g &= \mathrm{Prox}_{\alpha_k \lambda \|\cdot\|_p}([\boldsymbol{x}_k - \alpha_k \nabla f_{\mathcal{B}_k}(\boldsymbol{x}_k)]_g) \\
&= \left[I - \mathrm{Proj}^e_{\mathcal{B}(\|\cdot\|_{p'}, \alpha_k \lambda)}\right][\boldsymbol{x}_k - \alpha_k \nabla f_{\mathcal{B}_k}(\boldsymbol{x}_k)]_g \\
&= [\boldsymbol{x}_k - \alpha_k \nabla f_{\mathcal{B}_k}(\boldsymbol{x}_k)]_g - [\boldsymbol{x}_k - \alpha_k \nabla f_{\mathcal{B}_k}(\boldsymbol{x}_k)]_g = 0,
\end{aligned} \tag{82}$$

consequently $\mathcal{I}^0(\boldsymbol{x}^*) \subseteq \mathcal{I}^0(\boldsymbol{x}_{k+1})$, which completes the proof. $\square$

Now we establish the group-sparsity identification of HSPG as Theorem 2.

**Proof of Theorem 2:**

Suppose $\|\boldsymbol{x}_k - \boldsymbol{x}^*\| \leq \frac{2\alpha_k\delta_3}{1-\epsilon+\alpha_k L}$. There is nothing to prove if $g \in \mathcal{I}^0(\boldsymbol{x}^*)\bigcap\mathcal{I}^0(\boldsymbol{x}_k)$. For $g \in \mathcal{I}^0(\boldsymbol{x}^*)\bigcap\mathcal{I}^{\neq 0}(\boldsymbol{x}_k)$, we compute that

$$
\begin{aligned}
&[\boldsymbol{x}_k - \alpha_k\nabla\Psi_{\mathcal{B}_k}(\boldsymbol{x}_k)]_g^\top[\boldsymbol{x}_k]_g - \epsilon\|[\boldsymbol{x}_k]_g\|^2 \\
&= \|[\boldsymbol{x}_k]_g\|^2 - \alpha_k[\nabla\Psi_{\mathcal{B}_k}(\boldsymbol{x}_k)]_g^\top[\boldsymbol{x}_k]_g - \epsilon\|[\boldsymbol{x}_k]_g\|^2 \\
&= (1-\epsilon)\|[\boldsymbol{x}_k]_g\|^2 - \alpha_k\left([\nabla f_{\mathcal{B}_k}(\boldsymbol{x}_k)]_g + \lambda\frac{[\boldsymbol{x}_k]_g}{\|[\boldsymbol{x}_k]_g\|}\right)^\top[\boldsymbol{x}_k]_g \\
&= (1-\epsilon)\|[\boldsymbol{x}_k]_g\|^2 - \alpha_k[\nabla f_{\mathcal{B}_k}(\boldsymbol{x}_k)]_g^\top[\boldsymbol{x}_k]_g - \alpha_k\lambda\|[\boldsymbol{x}_k]_g\| \\
&\leq (1-\epsilon)\|[\boldsymbol{x}_k]_g\|^2 + \alpha_k\|[\nabla f_{\mathcal{B}_k}(\boldsymbol{x}_k)]_g\|\|[\boldsymbol{x}_k]_g\| - \alpha_k\lambda\|[\boldsymbol{x}_k]_g\| \\
&= \|[\boldsymbol{x}_k]_g\|\left\{(1-\epsilon)\|[\boldsymbol{x}_k]_g\| + \alpha_k\|[\nabla f_{\mathcal{B}_k}(\boldsymbol{x}_k)]_g\| - \alpha_k\lambda\right\}
\end{aligned}
\tag{83}
$$

By the Lipschitz continuity of $\nabla f$, we have that for each $g \in \mathcal{I}^0(\boldsymbol{x}^*)\bigcap\mathcal{I}^{\neq 0}(\boldsymbol{x}_k)$,

$$
\begin{aligned}
\|[\nabla f_{\mathcal{B}_k}(\boldsymbol{x}_k) - \nabla f_{\mathcal{B}_k}(\boldsymbol{x}^*)]_g\| &\leq L\|[\boldsymbol{x}_k - \boldsymbol{x}^*]_g\| = L\|[\boldsymbol{x}_k]_g\| \\
\|[\nabla f_{\mathcal{B}_k}(\boldsymbol{x}_k)]_g\| &\leq L\|[\boldsymbol{x}_k]_g\| + \|[\nabla f_{\mathcal{B}_k}(\boldsymbol{x}^*)]_g\|
\end{aligned}
\tag{84}
$$

Combining with the definition of $\delta_3$, which implies that $\|[\nabla f_{\mathcal{B}_k}(\boldsymbol{x}^*)]_g\| \leq \lambda - 2\delta_3$ that

$$
\|[\nabla f_{\mathcal{B}_k}(\boldsymbol{x}_k)]_g\| \leq L\|[\boldsymbol{x}_k]_g\| + \lambda - 2\delta_3
\tag{85}
$$

Hence combining with $\|[\boldsymbol{x}_k]_g\| \leq \frac{2\alpha_k\delta_3+\epsilon}{1+\alpha_k L}$, (83) can be further written as

$$
\begin{aligned}
&[\boldsymbol{x}_k - \alpha_k\nabla\Psi_{\mathcal{B}_k}(\boldsymbol{x}_k)]_g^\top[\boldsymbol{x}_k]_g - \epsilon\|[\boldsymbol{x}_k]_g\|^2 \\
&\leq \|[\boldsymbol{x}_k]_g\|\left\{(1-\epsilon)\|[\boldsymbol{x}_k]_g\| + \alpha_k\|[\nabla f_{\mathcal{B}_k}(\boldsymbol{x}_k)]_g\| - \alpha_k\lambda\right\} \\
&\leq \|[\boldsymbol{x}_k]_g\|\left\{(1-\epsilon)\|[\boldsymbol{x}_k]_g\| + \alpha_k L\|[\boldsymbol{x}_k]_g\| + \alpha_k\lambda - 2\alpha_k\delta_3 - \alpha_k\lambda\right\} \\
&= \|[\boldsymbol{x}_k]_g\|\left\{(1-\epsilon+\alpha_k L)\|[\boldsymbol{x}_k]_g\| - 2\alpha_k\delta_3\right\} \\
&\leq \|[\boldsymbol{x}_k]_g\|\left\{(1-\epsilon+\alpha_k L)\frac{2\alpha_k\delta_3}{1-\epsilon+\alpha_k L} - 2\alpha_k\delta_3\right\} \\
&= \|[\boldsymbol{x}_k]_g\|\left(2\alpha_k\delta_3 - 2\alpha_k\delta_3\right) = 0.
\end{aligned}
\tag{86}
$$

which shows that $[\boldsymbol{x}_k - \alpha_k\nabla\Psi_{\mathcal{B}_k}(\boldsymbol{x}_k)]_g^\top[\boldsymbol{x}_k]_g \leq \epsilon\|[\boldsymbol{x}_k]_g\|^2$. Hence the group projection operator is trigerred on $g$ to map the variables to zero, then $g \in \mathcal{I}^0(\boldsymbol{x}_{k+1})$, *i.e.*, $[\boldsymbol{x}_{k+1}]_g = 0$. Therefore, the group sparsity of $\boldsymbol{x}^*$ can be successfully identified by Half-Space Step, *i.e.*, $\mathcal{I}^0(\boldsymbol{x}^*) \subseteq \mathcal{I}^0(\boldsymbol{x}_{k+1})$.

### C.4 UPPER BOUND OF $N_{\mathcal{P}}$ UNDER STRONGLY CONVEXITY

**Proposition 2.** *Suppose the following conditions hold:*

- *(A1)* $\mathbb{E}[\nabla f_{\mathcal{B}_k}(\boldsymbol{x})] = \nabla f(\boldsymbol{x})$.

- *(A2) there exists a $\sigma > 0$ such that $\mathbb{E}_{\mathcal{B}}[\|\nabla f_{\mathcal{B}}(\boldsymbol{x}) - \nabla f(\boldsymbol{x})\|^2] \leq \sigma^2$ for any mini-batch $\mathcal{B}$.*

- *(A3) there exists a $\beta \in (0,1)$ such that $0 < \alpha_k < \frac{1-\beta}{L}$.*

- *(A4) $f$ is $\mu$-strongly convex.*

*Set the step-size $\alpha_k = \frac{1}{2\mu\beta k}$, $k_0 = \lceil\max\{1, \frac{1}{2\mu\beta}\}\rceil$. For any $\tau \in (0,1)$, there exists a $N_{\mathcal{P}} \in \mathbb{Z}^+$ such that $N_{\mathcal{P}} \geq \left\lceil\max\left\{\frac{8k_0\mathbb{E}[\|\boldsymbol{x}_{k_0}-\boldsymbol{x}^*\|^2]}{R^2\tau}, \frac{8\sigma^2\log(N_{\mathcal{P}}-1)}{\mu^2\beta^2 R^2\tau}\right\}\right\rceil$, such that performing Prox-SG $N_{\mathcal{P}}$ times yields*

$$
\|\boldsymbol{x}_{N_{\mathcal{P}}} - \boldsymbol{x}^*\| \leq R/2
\tag{87}
$$

*with probability at least $1 - \tau$.*

*Proof.* By the conditions (A1, A2, A3), Assumption 3.1 and Theorem 3.2 in Rosasco et al. (2019), we have for any $k \geq 2$,

$$\mathbb{E}[\|\boldsymbol{x}_k - \boldsymbol{x}^*\|^2] \leq \mathbb{E}[\|\boldsymbol{x}_{k_0} - \boldsymbol{x}^*\|^2]\left(\frac{k_0}{k}\right) + \frac{\sigma^2}{\mu^2\beta^2}\frac{\log(k-1)}{k}. \tag{88}$$

Let $\mathbb{E}[\|\boldsymbol{x}_{k_0} - \boldsymbol{x}^*\|^2] = s_{k_0}$. For any $\tau \in (0,1)$, there exists a $N_{\mathcal{P}} \in \mathbb{Z}^+$ satisfying

$$N_{\mathcal{P}} \geq \left\lceil \max\left\{\frac{8k_0 s_{k_0}}{R^2\tau}, \ \frac{8\sigma^2\log(N_{\mathcal{P}}-1)}{\mu^2\beta^2 R^2\tau}\right\}\right\rceil, \tag{89}$$

we have

$$\mathbb{E}[\|\boldsymbol{x}_{N_{\mathcal{P}}} - \boldsymbol{x}^*\|^2] \leq \frac{R^2\tau}{4}. \tag{90}$$

Therefore, by Markov inequality, we have that

$$\|\boldsymbol{x}_{N_{\mathcal{P}}} - \boldsymbol{x}^*\|^2 \leq \frac{R^2}{4} \ \Leftrightarrow \ \|\boldsymbol{x}_{N_{\mathcal{P}}} - \boldsymbol{x}^*\| \leq \frac{R}{2} \tag{91}$$

holds with probability at least $1 - \tau$.

$\square$

# D    ADDITIONAL NUMERICAL EXPERIMENTS

In this section, we provide additional numerical experiments to *(i)* demonstrate the validness of group sparsity identification of HSPG; *(ii)* provide comprehensive comparison to Prox-SG, RDA and Prox-SVRG on benchmark convex problems; and *(iii)* describe more details regarding our non-convex deep learning experiments shown in the main body.

## D.1    LINEAR REGRESSION ON SYNTHETIC DATA

We first numerically validate the proposed HSPG on group sparsity identification by linear regression problems with $\ell_1/\ell_2$ regularizations using synthetic data. Consider a data matrix $A \in \mathbb{R}^{N \times n}$ consisting of $N$ instances and the target variable $\boldsymbol{y} \in \mathbb{R}^N$, we are interested in the following problem:

$$\underset{\boldsymbol{x}\in\mathbb{R}^n}{\text{minimize}} \ \frac{1}{2N}\|A\boldsymbol{x} - \boldsymbol{y}\|^2 + \lambda\sum_{g\in\mathcal{G}}\|[\boldsymbol{x}]_g\|. \tag{92}$$

Our goal is to empirically show that HSPG is able to identify the ground truth zero groups with synthetic data. We conduct the experiments as follows: *(i)* generate the data matrix $A$ whose elements are uniformly distributed among $[-1, 1]$; *(ii)* generate a vector $\boldsymbol{x}^*$ working as the ground truth solution, where the elements are uniformly distributed among $[-1, 1]$ and the coordinates are equally divided into 10 groups ($|\mathcal{G}| = 10$); *(iii)* randomly set a number of groups of $\boldsymbol{x}^*$ to be 0 according to a pre-specified group sparsity ratio; *(iv)* compute the target variable $\boldsymbol{y} = A\boldsymbol{x}^*$; (v) solve the above problem (92) for $\boldsymbol{x}$ with $A$ and $\boldsymbol{y}$ only, and then evaluate the Intersection over Union (IoU) with respect to the identities of the zero groups between the computed solution estimate $\hat{\boldsymbol{x}}$ by HSPG and the ground truth $\boldsymbol{x}^*$.

We test HSPG on (92) under different problem settings. For a slim matrix $A$ where $N \geq n$, we test with various group sparsity ratios among $\{0.1, 0.3, 0.5, 0.7, 0.9\}$, and for a fat matrix $A$ where $N < n$, we only test with a certain group sparsity value since a recovery of $\boldsymbol{x}^*$ requires that the number of non-zero elements in $\boldsymbol{x}^*$ is bounded by $N$. Throughout the experiments, we set $\lambda$ to be $100/N$, the mini-batch size $|\mathcal{B}|$ to be 64, step size $\alpha_k$ to be 0.1 (constant), and fine-tune $\epsilon$ per problem. Based on a similar statistical test on objective function stationarity (Zhang et al., 2020), we switch to Half-Space Step roughly after 30 epoches. Table 2 shows that under each setting, the proposed HSPG correctly identifies the groups of zeros as indicated by IoU($\hat{\boldsymbol{x}}, \boldsymbol{x}^*$) = 1.0, which is a strong evidence to show the correctness of group sparsity identification of HSPG.

Table 2: Linear regression problem settings and IoU of the recovered solutions by HSPG.

|  | $N$ | $n$ | Group sparsity ratio of $\boldsymbol{x}^*$ | IoU$(\hat{x}, x^*)$ |
|---|---|---|---|---|
| | 10000 | 1000 | $\{0.1, 0.3, 0.5, 0.7, 0.9\}$ | 1.0 |
| Slim $A$ | 10000 | 2000 | $\{0.1, 0.3, 0.5, 0.7, 0.9\}$ | 1.0 |
| | 10000 | 3000 | $\{0.1, 0.3, 0.5, 0.7, 0.9\}$ | 1.0 |
| | 10000 | 4000 | $\{0.1, 0.3, 0.5, 0.7, 0.9\}$ | 1.0 |
| | 200 | 1000 | 0.9 | 1.0 |
| Fat $A$ | 300 | 1000 | 0.8 | 1.0 |
| | 400 | 1000 | 0.7 | 1.0 |
| | 500 | 1000 | 0.6 | 1.0 |

### D.2 LOGISTIC REGRESSION

We then focus on the benchmark convex logistic regression problem with the mixed $\ell_1/\ell_2$-regularization given $N$ examples $(\boldsymbol{d}_1, l_1), \cdots, (\boldsymbol{d}_N, l_N)$ where $\boldsymbol{d}_i \in \mathbb{R}^n$ and $l_i \in \{-1, 1\}$ with the form

$$\underset{(\boldsymbol{x};b) \in \mathbb{R}^{n+1}}{\text{minimize}} \frac{1}{N} \sum_{i=1}^{N} \log(1 + e^{-l_i(\boldsymbol{x}^T \boldsymbol{d}_i + b)}) + \lambda \sum_{g \in \mathcal{G}} \|[\boldsymbol{x}]_g\|, \tag{93}$$

for binary classification with a bias $b \in \mathbb{R}$. We set the regularization parameter $\lambda$ as $100/N$ throughout the experiments since it yields high sparse solutions and low object value $f$'s, equally decompose the variables into 10 groups to form $\mathcal{G}$, and test problem (93) on 8 standard publicly available large-scale datasets from LIBSVM repository (Chang & Lin, 2011) as summarized in Table 3. All convex experiments are conducted on a 64-bit operating system with an Intel(R) Core(TM) i7-7700K CPU @ 4.20 GHz and 32 GB random-access memory.

We run the solvers with a maximum number of epochs as 60. The mini-batch size $|\mathcal{B}|$ is set to be $\min\{256, \lceil 0.01N \rceil\}$ similarly to (Yang et al., 2019). The step size $\alpha_k$ setting follows [Section 4](Xiao & Zhang, 2014). Particularly, we first compute a Lipschitz constant $L$ as $\max_i \|\boldsymbol{d}_i\|^2 / 4$, then fine tune and select constant $\alpha_k \equiv \alpha = 1/L$ to Prox-SG and Prox-SVRG since it exhibits the best results. For RDA, the step size parameter $\gamma$ is fined tuned as the one with the best performance among all powers of 10. For HSPG, we set $\alpha_k$ as the same as Prox-SG and Prox-SVRG in practice. We set $N_{\mathcal{P}}$ as $30N/|\mathcal{B}|$ such that Half-Space Step is triggered after employing Prox-SG Step 30 epochs similarly to Appendix D.1, and the control parameter $\epsilon$ in (9) as 0.05. We select two $\epsilon$'s as 0 and 0.05. The final objective value $\Psi$ and $f$, and group sparsity in the solutions are reported in Table 4-6, where we mark the best values as bold to facilitate the comparison. Furthermore, Figure 3 plots the relative runtime of these solvers for each dataset, scaled by the runtime of the most time-consuming solver.

Table 6 shows that our HSPG is definitely the best solver on exploring the group sparsity of the solutions. In fact, HSPG under $\epsilon = 0.05$ performs all the best except *ijcnn1*. Prox-SVRG is the second best solver on group sparsity exploration, which demonstrates that the variance reduction techniques works well in convex setting to promote sparsity, but not in non-convex settings. HSPG under $\epsilon = 0$ performs much better than Prox-SG which matches the better sparsity recovery property of HSPG as stated in Theorem 2 even under $\epsilon$ as 0. Moreover, as shown in Table 4 and 5, we observe that all solvers perform quite competitively in terms of final objective values (round up to 3 decimals) except RDA, which demonstrates that HSPG reaches comparable convergence as Prox-SG and Prox-SVRG in practice. Finally, Figure 3 indicates that Prox-SG, RDA and HSPG have similar computational cost to proceed, except Prox-SVRG due to its periodical full gradient computation.

Table 3: Summary of datasets.

| Dataset | N | n | Attribute | Dataset | N | n | Attribute |
|---|---|---|---|---|---|---|---|
| a9a | 32561 | 123 | binary $\{0, 1\}$ | news20 | 19996 | 1355191 | unit-length |
| higgs | 11000000 | 28 | real $[-3, 41]$ | real-sim | 72309 | 20958 | real $[0, 1]$ |
| ijcnn1 | 49990 | 22 | real $[-1, 1]$ | url_combined | 2396130 | 3231961 | real $[-4, 9]$ |
| kdda | 8407752 | 20216830 | real $[-1, 4]$ | w8a | 49749 | 300 | binary $\{0, 1\}$ |

Table 4: Final objective values $\Psi$ for tested algorithms on convex problems.

| Dataset | Prox-SG | RDA | Prox-SVRG | HSPG | |
| --- | --- | --- | --- | --- | --- |
| | | | | $\epsilon$ as 0 | $\epsilon$ as 0.05 |
| a9a | **0.355** | 0.359 | **0.355** | **0.355** | **0.355** |
| higgs | **0.357** | 0.360 | 0.365 | 0.358 | 0.358 |
| ijcnn1 | **0.248** | 0.278 | **0.248** | **0.248** | **0.248** |
| kdda | **0.103** | 0.124 | **0.103** | **0.103** | **0.103** |
| news20 | **0.538** | 0.693 | **0.538** | **0.538** | **0.538** |
| real-sim | **0.242** | 0.666 | 0.244 | **0.242** | **0.242** |
| url_combined | 0.397 | 0.579 | **0.391** | 0.405 | 0.405 |
| w8a | **0.110** | 0.111 | 0.112 | **0.110** | **0.110** |

Table 5: Final objective values $f$ for tested algorithms on convex problems.

| Dataset | Prox-SG | RDA | Prox-SVRG | HSPG | |
| --- | --- | --- | --- | --- | --- |
| | | | | $\epsilon$ as 0 | $\epsilon$ as 0.05 |
| a9a | **0.329** | 0.338 | **0.329** | **0.329** | **0.329** |
| higgs | **0.357** | 0.360 | 0.365 | 0.358 | 0.358 |
| ijcnn1 | **0.213** | 0.270 | **0.213** | **0.213** | 0.214 |
| kdda | **0.103** | 0.124 | **0.103** | **0.103** | **0.103** |
| news20 | 0.373 | 0.693 | 0.381 | **0.372** | **0.372** |
| real-sim | **0.148** | 0.665 | 0.159 | **0.148** | **0.148** |
| url_combined | 0.397 | 0.579 | **0.391** | 0.405 | 0.405 |
| w8a | **0.089** | 0.098 | 0.091 | **0.089** | **0.089** |

Table 6: Group sparsity for tested algorithms on convex problems.

| Dataset | Prox-SG | RDA | Prox-SVRG | HSPG | |
| --- | --- | --- | --- | --- | --- |
| | | | | $\epsilon$ as 0 | $\epsilon$ as 0.05 |
| a9a | 20% | **30%** | **30%** | **30%** | **30%** |
| higgs | 0% | 10% | 0% | 0% | **30%** |
| ijcnn1 | 50% | **70%** | 60% | 60% | 60% |
| kdda | 0% | 0% | 0% | 0% | **80%** |
| news20 | 20% | 80% | **90%** | 80% | **90%** |
| real-sim | 0% | 0% | **80%** | 0% | **80%** |
| url_combined | 0% | 0% | 0% | 0% | **90%** |
| w8a | **0%** | **0%** | **0%** | **0%** | **0%** |

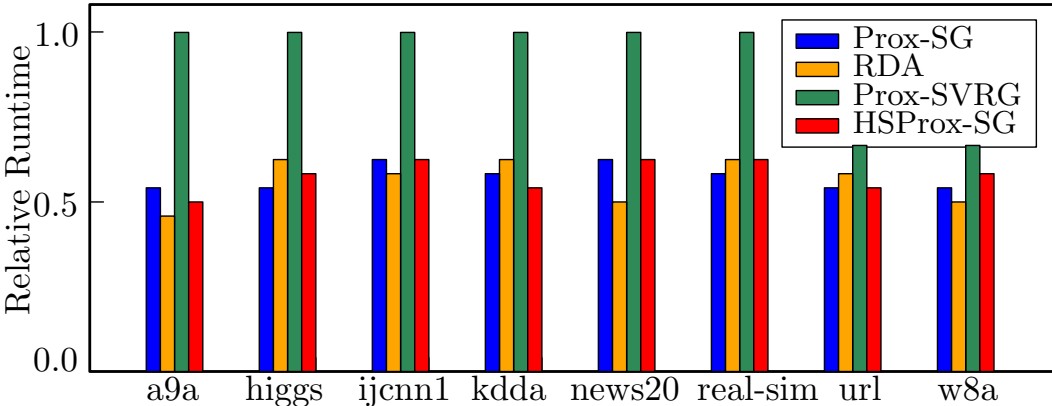

Figure 3: Relative runtime.

## D.3 DEEP LEARNING EXPERIMENTS

We conduct all deep learning experiments on one GeForce GTX 1080 Ti GPU, and describe how to fine-tune the control parameter $\epsilon$ in (9) in details.

**Fine-tune $\lambda$:** The $\lambda$ balances the sparsity level of the exact optimal solution and the bias of model estimation. Larger $\lambda$ encourages higher sparsity but may hurt the objective function. Therefore, in order to obtain solution of both high group sparsity and low objective value, we need to fine-tune $\lambda$ carefully. In our experiments, we iterate $\lambda$ through all powers of 10 from $10^{-5}$ to $10^{-1}$ by proceeding

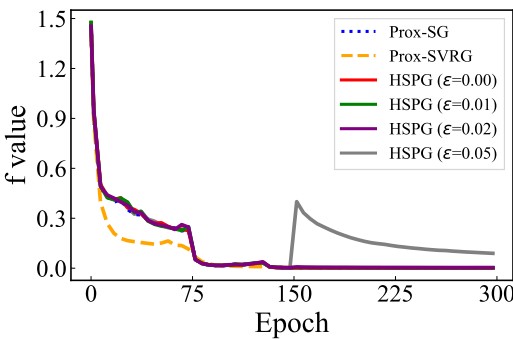

Figure 4: Evolution of $f$ value on ResNet18 with CIFAR10.

Prox-SG, and pick up the largest $\lambda$ which has the same level of test accuracy to the model trained without any regularization.

**Fine-tune $\epsilon$:** According to Theorem 2, a larger $\epsilon$ results in a faster group sparsity identification, while by Lemma 1 on the other hand too large $\epsilon$ may cause a significant regression on the target objective $\Psi$ value, *i.e.*, the $\Psi$ value increases a lot. Hence, in our experiments, from the point of view of optimization, we search a proper $\epsilon$ in the following ways: start from $\epsilon = 0.0$ and the models trained by employing $N_{\mathcal{P}}$ Prox-SG Steps, incrementally increase $\epsilon$ by 0.01 and check if the $\Psi$ on the first Half-Space Step has an obvious increase, then accept the largest $\epsilon$ without regression on $\Psi$ as our fine tuned $\epsilon$ shown in the main body of the paper. Particularly, the fine tuned $\epsilon$'s equal to 0.03, 0.05, 0.02 and 0.02 for VGG16 with CIFAR10, VGG16 with Fashion-MNIST, ResNet18 with CIFAR10 and ResNet18 with Fashion-MNIST respectively. Note from the perspective of different applications, there are different criterions to fine tune $\epsilon$, *i.e.*, for model compression, we may accept $\epsilon$ based on the validation accuracy regression to reach higher group sparsity.

**Final $f$ comparison:** Additionally, we also report the final $f$ comparison in Table 7 and its evolution on ResNet18 with CIFAR10 in Figure 4, where we can see that all tested algorithms can achieve competitive $f$ values as they do in convex settings. And the evolution of $f$ is similar to that of $\Psi$, *i.e.*, the raw objective $f$ generally monotonically decreases for small $\epsilon = 0$ to 0.02, and experiences a mild pulse after switch to Half-Space Step for larger $\epsilon$, *e.g.*, 0.05, which matches Lemma 1.

Table 7: Final objective values $f$ for tested algorithms on non-convex problems.

| Backbone | Dataset | Prox-SG | Prox-SVRG | HSPG $\epsilon$ as 0 | HSPG fine tuned $\epsilon$ |
|---|---|---|---|---|---|
| VGG16 | CIFAR10 | 0.010 | 0.036 | 0.010 | **0.009** |
|  | Fashion-MNIST | 0.181 | **0.165** | 0.181 | 0.182 |
| ResNet18 | CIFAR10 | **0.001** | 0.002 | **0.001** | 0.004 |
|  | Fashion-MNIST | 0.006 | 0.008 | **0.005** | 0.010 |
| MobileNetV1 | CIFAR10 | **0.021** | 0.031 | **0.021** | 0.031 |
|  | Fashion-MNIST | 0.074 | **0.057** | 0.074 | 0.088 |

