# OpenReview forum: "A Half-Space Stochastic Projected Gradient Method for Group Sparsity Regularization"
_ICLR.cc/2021/Conference — Reject_

### Official Review · AnonReviewer3 · 2020-10-26
**A good post processing trick to enhance group sparsity, but the usage of the group sparsity structure is unclear**

**Rating:** 5
**Confidence:** 3

**Review:**

In summary, this paper proposes a post-processing algorithm on the estimator obtained by the usual proximal Stochastic gradient method. This leads to an estimator with enhanced group sparsity without the sacrifice of accuracy.

My major concern is how to use such a group sparsity result? We end up with a more group sparse estimator, which is good. But I feel like that is not the end of the story. In a deep neural network, the group sparsity seems not the major point people care about. My hope is that the enhanced group sparsity can be used to guide maybe the design of the neural network structure, or at least provide some better understandings of the model. For example, if we always see that some groups of filters are inactive, then we may modify the neural network structure accordingly, etc. In all, my understanding is that the group sparsity could be an intermediate result that can be further analyzed and used to improve the design of the model, instead of being the final goal itself.

If we start with different initialization (probably just slightly different), then will we end up with the same final estimator or at least the same group sparsity results?

After obtaining the final estimator with group sparsity, can we refit the model on the active group index only? Will this improve the performance?

---

> ### Author Response · Authors · 2020-11-16
> **Author response for AnonReviewer3**
>
> Thank you for your thoughtful and detailed response. We have prepared a response, and modified the manuscript, to each of your points.
>
> * **Q1: My major concern is how to use such a group sparsity result? We end up with a more group sparse estimator, which is good. But I feel like that is not the end of the story. In a deep neural network, the group sparsity seems not the major point people care about. My hope is that the enhanced group sparsity can be used to guide maybe the design of the neural network structure, or at least provide some better understandings of the model. For example, if we always see that some groups of filters are inactive, then we may modify the neural network structure accordingly, etc. In all, my understanding is that the group sparsity could be an intermediate result that can be further analyzed and used to improve the design of the model, instead of being the final goal itself.**
>
>    A1: This is a great comment. In this paper, we focus on the optimization aspect to break the bottleneck of the group sparsity exploration which limits the usage of structured sparsity in neural network. Particularly, to solve group sparsity regularization optimization problem, we propose a novel HSPG supported by theoretical guarantee and numerical experiments which completes the story and largely tackle the limitations of existing algorithms on sparsity identification.
>
>    We definitely agree with you that one of the ultimate downstream applications of group sparsity is to construct better neural network architectures, e.g., removing redundancy for network compression by directly filtering out hidden structures entirely. However, to realize various downstream applications of group sparsity, we need to at first study how to solve the group-sparsity regularization problem effectively in the view of optimization as we have done in this paper, then exhaustively study the potential usage in neural network  which is out of the scope of optimization and can be treated as a separate future work.
>
>    Please see our more detailed consideration regarding structured learning in neural networks in a separate response if interested!
>
>
> * **Q2: If we start with different initialization (probably just slightly different), then will we end up with the same final estimator or at least the same group sparsity results?**
>
>   A2: Since the group-sparsity optimization problem in deep learning is non-convex and may have a lot of sparse stationary points. If we start from a different initial iterate, the HSPG may not converge to the same local sparse minimizer, and may end up on some other sparse minimizers. In our numerical experiments, we did not fix the random seed, but proceeded each experiment at least three times, and got consistent numerical results, e.g., similar final objective function, group sparsity level and validation accuracy. We have added a more detailed experimental setting in the Appendix D.3 of the revision.
>
> * **Q3: After obtaining the final estimator with group sparsity, can we refit the model on the active group index only? Will this improve the performance?**
>
>  A3: We can further refit on the model on the active group index only but there is typically no need for HSPG. Particularly,
>  	* **HSPG**: But for HSPG, there is typically no need to further refit the model on the active groups as (i) the model computed by HSPG does not regress on the objective convergence, which empirically implies no regression on the validation accuracy; (ii) if the group is defined properly, the inactive (zero) groups, e.g., the kernel of convolutional layer for fully CNNs (no residual or other special structures), can be directly removed, and the remaining trimmed model of active (nonzero) groups of parameters can proceed the same inference as before trimming without any refitting.
>  	* **Others**: For other optimizers, especially the ones equipped with simple truncation to filter out redundancy while dropped the accuracy a lot, the accuracy performance can improve by further retraining on the active groups, but it acquires additional steps, engineering efforts, and modifications on original optimization algorithms which is less attractive and user-friendly compared to HSPG which does not require these additional manipulations.
>
>  We have involved the above discussion into the Page 8 of the revision.

---

> > ### Author Response · Authors · 2020-11-16
> > **Supplementary discussion regarding structured learning in deep learning and why our work is important**
> >
> > * **The overview of structured sparsity in deep learning.** We notice that the structured sparsity is rarely well explored ("not the major point people care about") in deep learning for its potential natural applications, e.g., compressing and searching optimal neural network architecture, which is quite **unusual**, since the structured sparsity is widely used in many analogous applications, e.g., feature engineering in classical machine learning to filter out model redundancy by leveraging the distribution of zero entries on the computed weight parameters.
> >
> > * **The structured sparsity is solved by group sparsity.**  Remark here that as discussed in the Introduction of the main paper, the general structured sparsity problem are typically solved by converting into equivalent disjoint group sparsity problem in classical machine learning.
> >
> > * **Why the structured sparsity is rarely used in deep learning?** Based on our investigation and exploration, the limited usage of structured sparsity in deep learning is largely because that the existing classical stochastic proximal methods can not explore group sparse solution well, although they typically converge well on objective function value as revealed in this paper. Therefore, how to explore group sparsity effectively in stochastic learning is an important open problem which is crucial for the potential applications of structured sparsity in deep learning.
> >
> > * **Our HSPG breaks the bottleneck in the structured sparsity optimization.** Based on the above background, in order to extend structured sparsity into varying non-convex deep learning applications, the foremost step is to establish an effective stochastic optimization algorithm to overcome the limitation of the existing stochastic algorithm on the group sparsity identification. Therefore, in this paper, we study the group sparsity problem in the manner of optimization and propose a fresh and unique half-space method, HSPG, to explore group sparsity much more effectively than the others. The proposed HSPG can serve as the fundamental optimization algorithm to further support varying structured learning tasks in neural networks.
> >
> > * **Future work: study the applications of structured sparsity in deep learning based on HSPG.** As the next step, we are working on an application-track future paper to leverage the group sparsity into various potential deep learning applications, e.g., encoding the various structures of networks into groups to search an optimal model architecture by filtering out redundant structures by HSPG. We remark here that the application study of structured sparsity in deep learning is non-trivial and definitely worth a separate paper to dive in depth and explore exhaustively. Particularly, how to define the most proper group structure to different neural network architectures to formulate group sparsity regularization optimization problem is a valuable open problem. For example, for CNNs, we select weight parameters for each kernel in every convolution layer in the paper, while for CNNs with special structures, e.g., residual block and Sepconv, the definition of group may require special attentions. For RNNs, a standard way to define group may be choosing either row or column of weight matrices to construct low-rank weights, but how about they integrated with skip-connection, bi-direction and attention?  Furthermore, we expect that for graphical neural network and transformer, the study of structured sparsity may be even more complicated because of hierarchy. Therefore, we have chosen to not include the detailed application discussion of group sparsity into deep learning, but leave it as a separate future work.
> >
> > We thank again for the insightful comment and have included a Conclusion and Future Work section in the revision.

---

### Official Review · AnonReviewer1 · 2020-10-28
**Review of "A Half-Space Stochastic Projected Gradient Method for Group Sparsity Regularization"**

**Rating:** 5
**Confidence:** 2

**Review:**

The paper studies how to solve a class of group sparsity regularized minimization problems. In particular, a half-space stochastic projected gradient (HSPG) method is proposed, which is based on the Prox-SG and a new half-space step that promotes group sparsity. This step is to decompose the feasible space and then perform group projection. Convergence analysis is provided, together with the theoretical discussion that HSPG has looser requirements to identify the sparsity patter than Prox-SG. Numerical experiments on the DCNNs based image classification shows the proposed method achieves the state-of-the-art performance in terms of accuracy. The work looks interesting with wide applications, especially in deep neural networks. However, the novelty is incremental and limited.

1. There are some places where the notation is confusing. Vectors and scalars are constantly not distinguished.
2. Practical guidance on the selection of the parameters $\lambda$ and $\varepsilon$ could be provided.
3. In the numerical experiments, comparison of computational complexity and running time for the listed methods is not provided. Discussions on the group sparsity level and noise robustness could be included.

---

> ### Author Response · Authors · 2020-11-16
> **Author response for AnonReviewer1**
>
> We  thank  AnonReviewer1  for  the  elaborate  reviews  and  suggestions  which  are  helpful  for  us  to improve the paper.  Incorporating your suggestions has enhanced our work, and we hope the revision can improve the assessment of our paper. Below, please find the referee’s comments duplicated for ease of reference, along with our responses.
>
> * **Q1: The novelty is incremental and limited.**
>
>  A1: We kindly ask the reviewer to reconsider the novelty of this work as we are confident that the novelty is significant, especially our proposed half-space step to effectively explore group sparsity. In general, the proposed algorithm is
>    * **Unique**: To the best of knowledge, there is no similar group sparsity exploration method.
>    * **Fresh**:  Our method solve the (group) sparsity problem in a brand new way via a Half-Space Step, while the main-stream works on (group) sparsity have focused on using proximal method which is non-effective for group sparsity exploration.
>    * **Interesting / Good**: As other reviewers commented.
>
> * **Q2:  There are some places where the notation is confusing.  Vectors and scalars are constantly not distinguished.**
>
>    A2: That is a great suggestion.  We have distinguished the vectors and scalars via **bold** and non-bold.
>
> * **Q3:  Practical guidance on the selection of the parameters $\lambda$ and $\epsilon$ could be provided.**
>
>     A3: That is a good suggestion.
>
>     * For $\epsilon$, we have described how to fine-tune in the numerical experiments in the Appendix D.3, which is generally selected as the largest value without regression on objective function value when switch to Half-Space step, as larger $\epsilon$ results in more aggressive sparsity promotion while may hurt convergence.
>     * For $\lambda$, in the Appendix D.3 of the revision,  we have highlighted that the $\lambda$ is fine-tuned to be the largest value to reach competitive generalization accuracy to the model trained without regularization along with our explanation.
>
> * **Q4: In the numerical experiments, comparison of computational complexity and running  time  for  the  listed  methods  is  not  provided. Discussions on the  group sparsity level and noise robustness could be included**
>
>   A4: We have provided the runtime comparison among the tested solvers on the extensible experiments in Figure 3, Appendix D.2. Regarding the discussion on the group sparsity and noise robustness, we are not sure if we understand it properly, but we will try our best to answer your question. If the noise refers to the stochastic nature, it depends on the mini-batch size. Theoretically, larger mini-batch typically benefits the group sparsity identification. In our numerical experiments, we selected common mini-batch size setting $|B|\equiv 128$ as other related literatures, e.g., 'A Stochastic Extra-Step Quasi-Newton Method for Nonsmooth Nonconvex Optimization'.

---

### Official Review · AnonReviewer4 · 2020-10-28
**interesting algorithm to promote sparsity**

**Rating:** 5
**Confidence:** 3

**Review:**

This paper proposed a new algorithm for the group sparsity regularization problem. They claim most existing algorithms, though return solutions with low objective function value, only give dense solutions and cannot effectively ensure the desired structured sparsity. The new technique requires an initialization that is closed to some truly sparse local minimum, which is achieved by running proximal gradient descent first. Then, they proposed a new half-space iterative step to force elements in specific groups exactly to zero. The authors also provide convergence analysis and numerical evidence for the newly proposed algorithm.

Comments:
1. It is not clear to me, in Theorem 1, how are the parameters depend on the confidence \tau? I am confused as it seems no parameter is explicitly dependent on \tau, so the convergence in Theorem is almost surely one? I skim the proof and find that dependence vanished on the page (appendix) 10, proof of Lemma 6. I don’t understand why (1+\theta) can be omitted. Please clarify.
2. Where is N_P defined? N_P is used almost everywhere, for example, in statement of Theorem 1 and Algorithm 1. I didn’t find the definition of it. I guess N_P := min{k: ||x_k - x^*|| <= R/2} and R is further constrained by 2\delta_1 - R > 0.
3. In Theorem 1 and Proposition 1, only asymptotic and polynomial bounds are given. No rate of convergence for either the initialization phase or the half-space projection phase.

---

> ### Author Response · Authors · 2020-11-16
> **Author response for AnonReviewer4**
>
> Thank you for your constructive comments and suggestions, which is helpful to improve our paper. Please see our following responses along with the corresponding revision!
>
> * **Q1: It is not clear to me, in Theorem 1, how are the parameters depend on the confidence $\tau$? I am confused as it seems no parameter is explicitly dependent on $\tau$, so the convergence in Theorem is almost surely one? I skim the proof and find that dependence vanished on the page (appendix) 10, proof of Lemma 6. I don’t understand why $(1+\theta)$ can be omitted. Please clarify.**
>
>   A1: This is good question. In short, in Theorem 1, the setting of $\alpha_k$ and mini-batch size $\mathcal{B}_k$ depends on $\tau$, while the dependence is  implicit without closed form, hence we did not include $\tau$.  Please see our more detailed explanations as a separate response if interested!
>
> * **Q2: Where is $N_P$ defined?**
>
>   A2: We have added the formal definition of $N_P$ on the page 3 in the revision.
>
> * **Q3: In Theorem 1 and Proposition 1, only asymptotic and polynomial bounds are given. No rate of convergence for either the initialization phase or the half-space projection phase.**
>
>   A3: This is a great suggestion. We have chosen not to include rate of convergence after careful consideration, since (i) the paper focuses on the rarely-explored group sparsity identification property in stochastic learning, but not the rate of convergence which has been very well explored; (ii) both stages are essentially basic stochastic gradient descent without any acceleration mechanisms, i.e., momentum etc., but equipped with proximal mapping and our proposed novel half-space projection respectively. For basic stochastic gradient descent, there have been well-established convergence rate results.
>
>    Therefore, we would prefer not to discuss the convergence rate in details as we may easily pump up the rate by incorporating some acceleration mechanism, e.g., employing Prox-SVRG instead of Prox-SG as the initialization stage or averaging gradients in the half-space step, while may distract the group sparsity exploration focus instead.
>
>    We have added a remark to summarize the above in Page 7 in the revision.

---

> > ### Author Response · Authors · 2020-11-16
> > **Further clarification regarding Q1/A1.**
> >
> > To further clarify, in order to prove the convergence of the second stage by Half-Space step, we require the starting iterate $x_{N_P}$ closed enough to the $x^*$, i.e., $||x_k-x^*||\leq R/2$. The reasons behind the closed-enough requirement is that if the current iterate of Half-Space step $x_k$ falls into $\tilde{\mathcal{X}}:=${$x:x\in\mathbb{R}^n, ||x-x^*||\leq R$}, then the following properties holds:
> >
> > * **Cover group support**: Lemma 3 indicates if the current iterate $x_k$ closed enough to the local minimizer $x^*$, i.e., $||x_k-x^*||\leq R$, then $x_k$ has already cover the supports (non-zero groups) of the solution.
> >
> > * **The local minimizer inhabits the constructed half-space reduced space**: Lemma 4 indicates $x^*$ inhabits the constructed reduced space $S_k$ if $x_k$ is closed enough, i.e., $||x_k-x^*||\leq R$. Hence, optimizing over the constructed reduced space can make progress to converge to $x^*$.
> >
> > * **Correctness of newly identified zero groups**: Lemma 5 further indicates if current iterate $x_k$ closed enough to $x^*$, i.e., $||x_k-x^*||\leq R$, and step size is sufficiently small, then every newly projected groups to zero by Half-Space step are definitely zeros on the solution as well.
> >
> > Based on the above points, if Half-Space step can guarantee every iterate closed enough to the $x^*$, then the convergence of objective, correct group sparsity identification and support recovery would be ultimately achieved. But due to the randomness and noise of gradient estimation, the intermediate iterates of Half-Space step may falls outside of $\tilde{\mathcal{X}}$. Therefore, we aim at obtaining the iterate sequence of Half-Space step inhabits $\tilde{\mathcal{X}}$ with high probability.
> >
> > To bound the probability of some iterates falling outside of $\tilde{\mathcal{X}}$, we initialize the second stage from iterate which satisfies $||{x}_{N_P}-x^*||\leq R/2$, but not $R$. Then we move on to bound the error on the distance between the iterate of Half-Space Step and $x^*$ caused by randomness in Lemma 6, followed by showing each individual iterate of Half-Space step falls into $\tilde{\mathcal{X}}$ with high probability in Lemma 7.
> >
> > To further ensure the whole sequence of iterates of Half-Space step inhabits in $\tilde{\mathcal{X}}$ with high probability as Lemma 8, we need to set step size $\alpha_k$ linearly decaying and mini-batch size $B_k$ linearly increasing. We remark here that the setting of $\alpha_k$ and $\mathcal{B}_k$ depend on the confidence parameter, which vanished. Since in equation (53) of Lemma 6, there is no explicit form of the series about $\alpha_k$ and $\mathcal{B}_k$. Hence for any confidence parameter, there exist big O settings of $\alpha_k=O(1/k)$ and $B_k=O(k)$ to ensure all iterates of Half-Space steps locate in $\tilde{\mathcal{X}}$ with high probability, while the dependency is implicit.

---

### Official Review · AnonReviewer2 · 2020-10-29
**The idea is interesting, but I have a concern.**

**Rating:** 6
**Confidence:** 3

**Review:**

[Summary]
This paper proposes a new method called Half-space Stochastic Projected Gradient (HSPG) to find a group sparse solution of regularized finite-sum problems. Theoretical analysis tries to show the sparsity identification guarantees. In experiments, the effectiveness of HSPG was verified on the classification tasks.

[Strength]
The idea behind the proposed method seems to be reasonable and interesting.

[Weakness]
A major concern is the correctness of the statements. In the equation (97) in the proof, the equation $E_B[e(x)] = 0$ is used essentially, and it is also stated in page 6. However, I think it does not hold because the proximal operator associated with sparse regularization is nonlinear. It may be probably difficult to fix this issue.

[Minor comment]
There is a missing reference. It is known that RDA has the superior ability to find a manifold structure of solutions as shown in the following paper.

S. Lee and S. J. Wright. Manifold identification in dual averaging for regularized stochastic online learning. JMLR, 2012.

[Improvement]
If there is a misunderstanding in my review, I'd appreciate it if you could mention them.

---

> ### Author Response · Authors · 2020-11-16
> **Author response for AnonReviewer2**
>
> Thank you very much for your extraordinary correction and detailed responses. We really appreciate the bias of stochastic proximal gradient which is crucial for us. We hope the revision  can improve the assessment of our paper.
>
> * **Q1:  A major concern is the correctness of the statements.  In the equation (97) in  the  proof,  the  equation  is  used  essentially,  and  it  is  also  stated  in  page  6. However, I think it does not hold because the proximal operator associated with sparse regularization is nonlinear.  It may be probably difficult to fix this issue.**
>
>   A1: This is a great catch!  We sincerely appreciate you for pointing it out.  We admit the assumption used for the proposition 1 indeed does not hold after careful validation, and we have fixed it in the revision.
>
>   * **Impact of the unbiased assumption**: Generally, this wrong assumption is used in Proposition 1, but does not affect the main theorems of HSPG, i.e., Theorem 1 and 2. As shown in the below, the revised proposition is mainly built on existing analysis results, so that it can not be considered as our novel contributions. Therefore, to emphasize our main contributions, we would like to remove this proposition from the main body, and add a fixed version into the Appendix C.4.
>
>   *  **Outline of Revised and fixed proposition**: Proposition 1 provides a sufficient (not necessary) condition to satisfy the $||x_k−x^∗||\leq R/2$ required from Theorem 1 by performing Prox-SG sufficiently many times with high prob-ability.  To fix the proposition,  in Appendix C.4, we have provided an alternative proposition  under  a  stronger  assumption,  i.e.,  strong  convexity,  compared  to  the PL condition.
>
>    Overall, to guarantee $||x_k−x^∗||\leq R/2$ with high probability, the general idea is to show that the expectation of $||x_k−x^∗||$ is upper bounded by some constant less than $R/2$, then apply Markov inequality to construct a high probability result.  To bound the expectation of $||x_k−x^*||$, under strong convexity, we notice that there exist some existing applicable results, e.g., Theorem 3.2 in 'Lorenzo Rosasco, Silvia Villa, and Bằng Công Vũ, Convergence of Stochastic Proximal Gradient Algorithm, 2019'. Therefore, we have obtained a similar upper bound of $N_P$ as previous.
>
> 	Many thanks again for the error correction!
>
> * **Q2:  There is a missing reference.  It is known that RDA has the superior ability to find a manifold structure of solutions.**
>
> 	A2: We have added a more complete literature review section, which includes new citation to the missing reference.

---

> > ### Comment · AnonReviewer2 · 2020-11-25
> > **Thank you for the reply.**
> >
> > Thank you for the reply.
> > The authors well addressed my main concern and I am convinced of the correctness of the revised version.
> > I have raised the score.

---

### Decision · Program_Chairs · 2021-01-07
**Final Decision**

**Decision:**

Reject

**Comment:**

The paper received four borderline reviews.  Overall, the manuscript has improved after the rebuttal (in particular, an issue in the convergence proof has been fixed), and a reviewer has increased his score to borderline accept. Yet, the paper did not convince the reviewers that the contribution was significant enough and none of the reviewer got enthusiastic about the paper. The main issue with the paper seems to be the unclear positioning between the optimization literature for stochastic composite optimization, the literature on support identification (e.g., Nutini, 2019), and the (more empirical) deep learning literature.

The paper postulates that the group-sparsity regularization is crucial for deep neural networks, which seems to be the main motivation of the paper. Yet, the experiments do not demonstrate any concrete consequence of better group sparsity, wether it is in terms of accuracy or interpretability.  If positioned in this literature, a comparison should be made with classical pruning approaches, where pruning occurs as an iterative procedure that is distinct from optimization. If positioned instead in the stochastic optimization literature, better analysis of the convergence rates should be provided; if positioned in the support identification literature, the paper should explain how the results compare to those of the literature (e.g., Nutini, 2019 and others). In other words, any point of view requires clarifications and additional discussions.

Besides,
   - the theoretical assumptions need to be discussed:  does the Lipschitz assumption holds for multilayer neural networks ? Certainly not for ReLu networks, but what can we say something useful, even with smooth activation functions?
   - the experimental setup needs more details. Reproducing the experiments with the current paper seems difficult; in particular, the choice of hyper-parameters is not crystal clear.

For these reasons, the area chair recommends to reject the paper, but encourages the authors to resubmit to a future venue while taking into account the previous comments.